# GWAS and ExWAS of blood mitochondrial DNA copy number identifies 71 loci and highlights a potential causal role in dementia

Michael Chong[1,2,3], Pedrum Mohammadi-Shemirani[2,3,4], Nicolas Perrot[3], Walter Nelson[5], Robert Morton[2,3], Sukrit Narula[3,6], Ricky Lali[3,6], Irfan Khan[2,3], Mohammad Khan[3,7], Conor Judge[3,8], Tafadzwa Machipisa[2,3,9,10], Nathan Cawte[3], Martin O'Donnell[3,8], Marie Pigeyre[3,7], Loubna Akhabir[3,7], Guillaume Paré[1,2,3,6,7]*

[1]Department of Biochemistry and Biomedical Sciences, McMaster University, Hamilton, Canada; [2]Department of Pathology and Molecular Medicine, McMaster University, Hamilton, Canada; [3]Population Health Research Institute, David Braley Cardiac, Vascular and Stroke Research Institute, Hamilton Health Sciences, Hamilton, Canada; [4]Thrombosis and Atherosclerosis Research Institute, Hamilton, Canada; [5]Centre for Data Science and Digital Health, Hamilton Health Sciences, Hamilton, Canada; [6]Department of Health Research Methods, Evidence, and Impact, McMaster University, Hamilton, Canada; [7]Department of Medicine, McMaster University, Michael G. DeGroote School of Medicine, Hamilton, Canada; [8]National University of Ireland, Galway, Galway, Ireland; [9]Department of Medicine, University of Cape Town & Groote Schuur Hospital, Cape Town, South Africa; [10]Hatter Institute for Cardiovascular Diseases Research in Africa (HICRA) & Cape Heart Institute (CHI), Department of Medicine, University of Cape Town, Cape Town, South Africa

*For correspondence: pareg@mcmaster.ca

## Abstract

**Background:** Mitochondrial DNA copy number (mtDNA-CN) is an accessible blood-based measurement believed to capture underlying mitochondrial (MT) function. The specific biological processes underpinning its regulation, and whether those processes are causative for disease, is an area of active investigation.

**Methods:** We developed a novel method for array-based mtDNA-CN estimation suitable for biobank-scale studies, called 'automatic mitochondrial copy (AutoMitoC).' We applied AutoMitoC to 395,781 UKBiobank study participants and performed genome- and exome-wide association studies, identifying novel common and rare genetic determinants. Finally, we performed two-sample Mendelian randomization to assess whether genetically low mtDNA-CN influenced select MT phenotypes.

**Results:** Overall, genetic analyses identified 71 loci for mtDNA-CN, which implicated several genes involved in rare mtDNA depletion disorders, deoxynucleoside triphosphate (dNTP) metabolism, and the MT central dogma. Rare variant analysis identified *SAMHD1* mutation carriers as having higher mtDNA-CN (beta = 0.23 SDs; 95% CI, 0.18–0.29; p=2.6 × 10⁻¹⁹), a potential therapeutic target for patients with mtDNA depletion disorders, but at increased risk of breast cancer (OR = 1.91; 95% CI, 1.52–2.40; p=2.7 × 10⁻⁸). Finally, Mendelian randomization analyses suggest a causal effect of low mtDNA-CN on dementia risk (OR = 1.94 per 1 SD decrease in mtDNA-CN; 95% CI, 1.55–2.32; p=7.5 × 10⁻⁴).

**Conclusions:** Altogether, our genetic findings indicate that mtDNA-CN is a complex biomarker reflecting specific MT processes related to mtDNA regulation, and that these processes are causally related to human diseases.

**Funding:** No funds supported this specific investigation. Awards and positions supporting authors include: Canadian Institutes of Health Research (CIHR) Frederick Banting and Charles Best Canada Graduate Scholarships Doctoral Award (MC, PM); CIHR Post-Doctoral Fellowship Award (RM); Wellcome Trust Grant number: 099313/B/12/A; Crasnow Travel Scholarship; Bongani Mayosi UCT-PHRI Scholarship 2019/2020 (TM); Wellcome Trust Health Research Board Irish Clinical Academic Training (ICAT) Programme Grant Number: 203930/B/16/Z (CJ); European Research Council COSIP Grant Number: 640580 (MO); E.J. Moran Campbell Internal Career Research Award (MP); CISCO Professorship in Integrated Health Systems and Canada Research Chair in Genetic and Molecular Epidemiology (GP)

## Editor's evaluation

This is an original human GWAS study that treats mitochondrial copy number variation as a trait, and investigates its genetic basis, as well as its association with (and possible causal role in) various human diseases, such as cancer and dementia. The study identifies 71 significant loci, show that these are significantly over-represented in a priori candidates, and argue convincingly that this could help us understand how mitochondrial copy number is regulated at a cellular level.

## Introduction

Mitochondria are semiautonomous organelles present in nearly every human cell that execute fundamental cellular processes including oxidative phosphorylation, calcium storage, and apoptotic signaling. Mitochondrial (MT) dysfunction has been implicated as the underlying cause for many human disorders based on mechanistic in vitro and in vivo studies (*Burbulla et al., 2017*; *Desdín-Micó et al., 2020*; *Sliter et al., 2018*). Complementary evidence comes from recent epidemiological studies that measure mitochondrial DNA copy number (mtDNA-CN), an MT-derived marker that can be conveniently measured from peripheral blood. Since mitochondria contain their own unique set of genomes that are distinct from the nuclear genome, the ratio of mtDNA to nuclear DNA molecules (mtDNA-CN) in a sample serves as an accessible marker of MT DNA abundance per cell (*Longchamps et al., 2020*). Indeed, observational studies suggest that individuals with lower mtDNA-CN are at higher risk of age-related complex diseases, such as coronary artery disease, sudden cardiac death, cardiomegaly, stroke, portal hypertension, and chronic kidney disease (*Tin et al., 2016*; *Ashar et al., 2017*; *Zhang et al., 2017*; *Hägg et al., 2021*). Conversely, higher mtDNA-CN levels have been associated with increased cancer incidence (*Kim et al., 2015*; *Hu et al., 2016*).

While previous studies demonstrate that mtDNA-CN is associated with various diseases, evidence suggests that it may also play a direct and causative role in human health and disease. For example, in cases of mtDNA depletion syndrome, wherein rare defects in nuclear genes responsible for replicating and/or maintaining mtDNA lead to deficient mtDNA-CN (*Gorman et al., 2016*), patients manifest with severe dysfunction of energy-dependent tissues (heart, brain, liver, and cardiac and skeletal muscles). So far, 19 genes have been reported to cause mtDNA depletion (*Oyston, 1998*). In addition to these rare monogenic syndromes, the importance of common genetic variation in regulating mtDNA-CN is an active area of research with approximately 50 common loci identified so far (*Cai et al., 2015*; *Guyatt et al., 2019*; *Longchamps, 2019*; *Hägg et al., 2021*). In contrast to marked drops in mtDNA-CN by 60–80% as seen in those with rare mtDNA depletion syndromes, the relevance of subtler perturbations in mtDNA-CN in disease risk remains to be determined (*Basel, 2020*).

Granted, the connection between blood mtDNA-CN and aspects of MT biology remains unclear with many studies showing only moderate correlation between mtDNA-CN and markers of MT function or biogenesis (*Frahm et al., 2005*; *Wachsmuth et al., 2016*). Further complicating the interpretation of epidemiological associations between blood mtDNA-CN and disease risk is the fact that (i) blood mtDNA-CN is strongly confounded by blood cell composition, particularly, platelets that are devoid of nuclei and that (ii) blood mtDNA-CN does not correlate well with mtDNA-CN measured in other tissues in which MT dysfunction may be more relevant (*Picard, 2021*). Accordingly, understanding

**eLife digest** Our cells are powered by small internal compartments known as mitochondria, which host several copies of their own 'mitochondrial' genome. Defects in these semi-autonomous structures are associated with a range of severe, and sometimes fatal conditions: easily checking the health of mitochondria through cheap, quick and non-invasive methods can therefore help to improve human health.

Measuring the concentration of mitochondrial DNA molecules in our blood cells can help to estimate the number of mitochondrial genome copies per cell, which in turn act as a proxy for the health of the compartment. In fact, having lower or higher concentration of mitochondrial DNA molecules is associated with diseases such as cancer, stroke, or cardiac conditions. However, current approaches to assess this biomarker are time and resource-intensive; they also do not work well across people with different ancestries, who have slightly different versions of mitochondrial genomes.

In response, Chong et al. developed a new method for estimating mitochondrial DNA concentration in blood samples. Called AutoMitoC, the automated pipeline is fast, easy to use, and can be used across ethnicities. Applying this method to nearly 400,000 individuals highlighted 71 genetic regions for which slight sequence differences were associated with changes in mitochondrial DNA concentration. Further investigation revealed that these regions contained genes that help to build, maintain, and organize mitochondrial DNA. In addition, the analyses yield preliminary evidence showing that lower concentration of mitochondrial DNA may be linked to a higher risk of dementia.

Overall, the work by Chong et al. demonstrates that AutoMitoC can be used to investigate how mitochondria are linked to health and disease in populations across the world, potentially paving the way for new therapeutic approaches.

the genetic determinants of blood mtDNA-CN may provide a better understanding of the etiological processes reflected by this poorly understood MT biomarker in the blood.

To interrogate mtDNA-CN as a potential determinant of human diseases and to better understand its biological relevance to mitochondria, we performed extensive genetic investigations in up to 395,781 participants from the UKBiobank study (*Sudlow et al., 2015*). We first developed and validated a novel method for biobank-scale mtDNA-CN investigations that leverages single nucleotide polymorphism (SNP) array intensities, called 'automatic mitochondrial copy (AutoMitoC).' Leveraging AutoMitoC-based mtDNA-CN estimates, we performed large-scale genome-wide association study (GWAS) and exome-wide association study (ExWAS) to identify common and rare genetic variants contributing to population-level variation in mtDNA-CN. Various analyses were then conducted to build on previous publications regarding the specific genes and pathways underlying mtDNA-CN regulation (*Cai et al., 2015*; *Guyatt et al., 2019*; *Longchamps, 2019*; *Hägg et al., 2021*). Finally, we applied Mendelian randomization analyses to assess potential causal relationships between mtDNA-CN and disease susceptibility.

## Results

### AutoMitoC: a streamlined method for array-based mtDNA-CN estimation

We built on an existing framework for processing normalized SNP probe intensities (log2ratio [L2R] values) from genetic arrays into mtDNA-CN estimates known as the 'MitoPipeline' (*Lane, 2014*) (Materials and Methods, *Figure 1*). The MitoPipeline yields mtDNA-CN estimates that correlate with direct qPCR measurements and has been successfully implemented in several epidemiological investigations (*Ashar et al., 2017*; *Zhang et al., 2017*; *Figure 1—figure supplements 1–2*). We developed a novel method, 'AutoMitoC,' which incorporates three amendments to facilitate large-scale investigations of mtDNA-CN (*Figure 1*). First, AutoMitoC replaces autosomal signal normalization of common variants with globally rare variants which negates the need for linkage disequilibrium pruning (Materials and Methods, *Figure 1—figure supplements 3–4*). As a result, this simplifies derivation of mtDNA-CN estimates in ethnically diverse cohorts by allowing for use of a single, universal variant set for normalization. Second, to detect potentially cross-hybridizing probes, we empirically assess

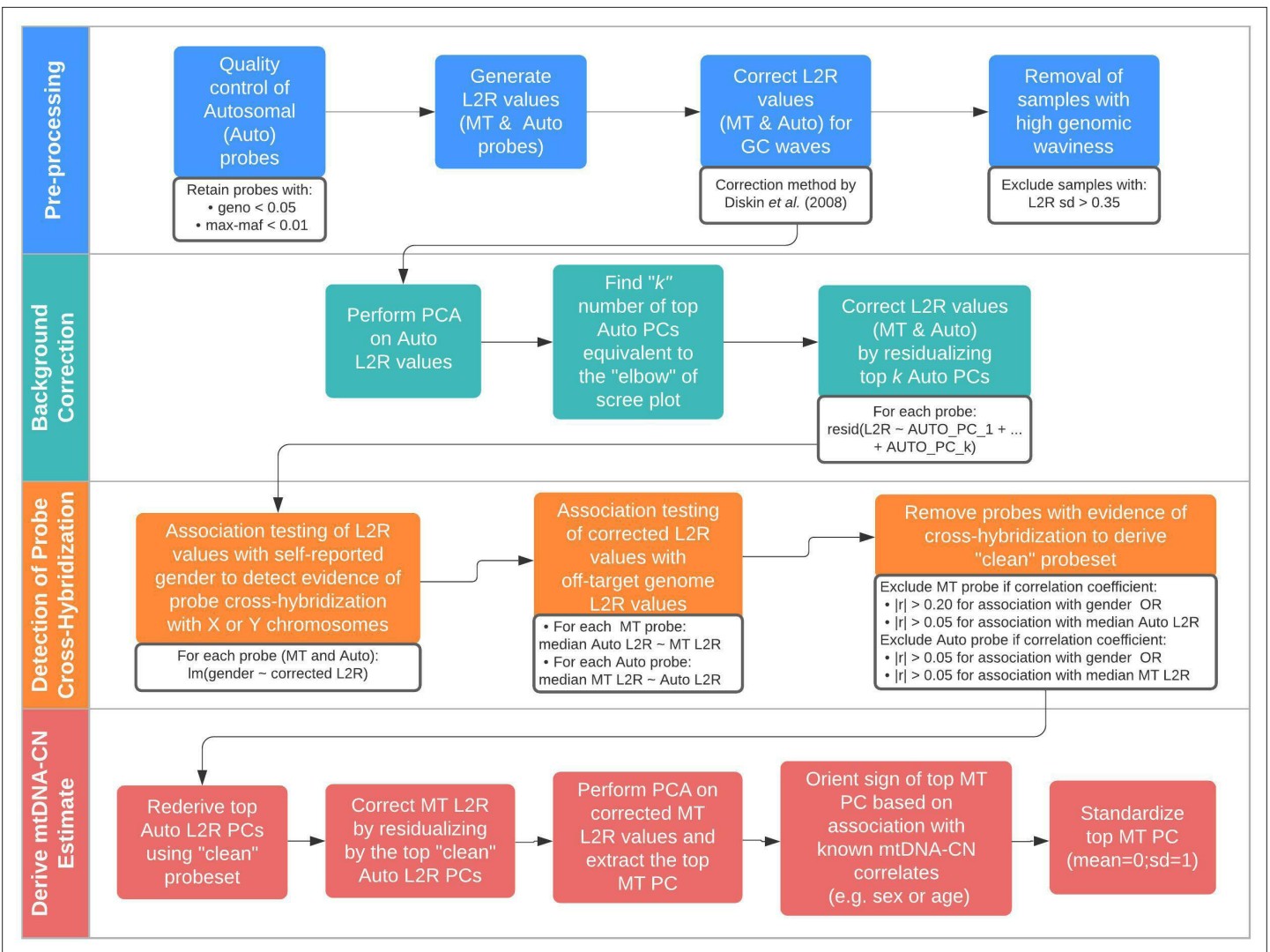

**Figure 1.** Schematic summary of the automatic mitochondrial copy (AutoMitoC) pipeline. The AutoMitoC pipeline is comprised of four major steps: (i) preprocessing, (ii) background correction, (iii) detection of probe cross-hybridization, and (iv) final derivation of mitochondrial DNA copy number (mtDNA-CN) estimates. First, preprocessing is simplified by restricting analysis of autosomal variants to those that have low minor allele frequency ( <0.01) and low genotype missingness ( <0.05). For probes passing quality control, MT and autosomal log2ratio (L2R) values undergo an initial correction for guanine cytosine (GC) waves using the method by **Diskin et al., 2008**. Samples exhibiting high genomic waviness post GC-correction (L2R SD >0.35) are removed. Second, background correction consists of performing principal component analysis of the autosomal probe L2R values and finding the top $k$ principal components (PCs) that correspond to the 'elbow' of the scree plot. In our case, ~70% variance in autosomal L2R values was explained by the top $k$ PCs in both UKbiobank and INTERSTROKE datasets. GC-corrected MT and L2R values are then further adjusted for the top autosomal PCs (representing technical background noise) by taking the residuals of the association between the L2R values versus the $k$ autosomal PCs. Third, we derive a 'clean' set of autosomal and MT probes without signs of off-target probe cross-hybridization by empirically testing the GC-corrected and background-corrected L2R values for association with either the sample medians of off-target genome L2R values or self-reported gender (to capture off-target hybridization to sex chromosomes). Fourth, using the 'clean' probeset, we repeat the autosomal background correction, extract the top MT PC as a crude measure of mtDNA-CN, change the sign of the MT PC according to association of the MT PC with known predictors of mtDNA-CN that are commonly reported (sex or age), and last, standardize the MT PC values as the final AutoMitoC estimate.

The online version of this article includes the following figure supplement(s) for figure 1:

**Figure supplement 1.** Intuition behind differentiation of genotypes and determination of mitochondrial DNA copy number (mtDNA-CN).

**Figure supplement 2.** Overview of the MitoPipeline (Source: http://genvisis.org/MitoPipeline/) (**Lane, 2014**).

**Figure supplement 3.** Minor allele frequency (MAF)-stratified analyses demonstrating utility of rare vs common autosomal variants for signal normalization.

**Figure supplement 4.** Distribution of log10 transformed coefficients of determination ($r^2$) from the association between autosomal probe intensities and median mitochondrial (MT) signal with (blue) or without (red) correction for background noise (i.e. 120 autosomal principal components [PCs]).

*Figure 1 continued on next page*

*Figure 1 continued*

**Figure supplement 5.** Validation of automatic mitochondrial copy in an ethnically diverse cohort with qPCR measurements.

**Figure supplement 6.** Bland Altman plots illustrating the extent of agreement between array and qPCR measurements.

the association of corrected probe signal intensities with off-target genome intensities (Materials and Methods, *Supplementary file 1* – Tab 1–2), rather than relying on sequence homology of probe sequences, which is not always available. Last, the primary estimate of MT signal is ascertained using principal component analysis ([PCA]; as opposed to using the median signal intensity of MT probes as per the MitoPipeline) which improves concordance of array-based mtDNA-CN estimates with those derived from alternative methods (Materials and Methods). A detailed description of the development of the AutoMitoC pipeline is provided in the Materials and Methods.

To benchmark performance of AutoMitoC, array-based mtDNA-CN estimates were compared to complementary measures of mtDNA-CN in two independent studies. First, array-based mtDNA-CN estimates were derived in a subset of 34,436 UKBiobank participants with available whole exome sequencing (WES) data. Reference mtDNA-CN estimates were derived from the proportion of WES reads aligned to the MT genome relative to the autosome (*Longchamps, 2019*). AutoMitoC estimates were significantly correlated with WES estimates ($r$ = 0.45; $p < 2.23 \times 10^{-308}$). Since WES data involves enrichment for nuclear coding genes and therefore could result in biased reference estimates for mtDNA-CN, we also performed an independent validation in an ethnically diverse study of 5791 participants where mtDNA-CN was measured using qPCR, the current gold standard assay (*Fazzini et al., 2018*). Indeed, we observed stronger correlation between AutoMitoC and qPCR-based estimates ($r$ = 0.64; $p < 2.23 \times 10^{-308}$; *Figure 1—figure supplement 5*). Furthermore, AutoMitoC demonstrated robust performance ($r \geq 0.53$) across all ethnic strata in the secondary validation cohort including Europeans (N = 2431), Latin Americans (N = 1704), Africans (N = 542), South East Asians (N = 471), South Asians (N = 186), and others (N = 360; ; *Figure 1—figure supplement 5*). Bland Altman plots also illustrate the extent of agreement between methods (*Figure 1—figure supplement 6*). For every ethnicity, 95% limits of agreement intervals were smaller than expected by chance. Last, while all analyses hitherto followed the MitoPipeline condition of requiring >40,000 autosomal variants for normalization, we observed comparable performance using even 1000 random very rare probes (MAF <0.001; $r$ = 0.60; $p < 5 \times 10^{-300}$) for signal normalization which reduced the runtime from several hours to less than 10 min for these 5791 samples running on 10 CPUs.

In the larger UKBiobank dataset of 395,781 with suitable array-based estimates, the distribution of mtDNA-CN was approximately normal (*Figure 2A*). To further verify that AutoMitoC-based estimates were indeed capturing mtDNA-CN, we performed association testing between mtDNA-CN values and known predictors including age, sex, ethnicity, and blood cell composition. Consistent with previous reports, every decade increase in age was associated with lower levels (beta = −0.08 SDs; 95% CI, –0.07 to –0.08; $p < 2.23 \times 10^{-308}$), and females had higher levels than males (beta = 0.12 SDs; 95% CI, 0.11–0.12; $p = 1.67 \times 10^{-299}$). MtDNA-CN levels also differed between ethnicities with South Asians (beta = −0.18; 95% CI, −0.16 to −0.21; $p = 2.93 \times 10^{-47}$) and Africans (beta = −0.18; 95% CI, −0.16 to −0.21; $p = 8.39 \times 10^{-48}$) having significantly lower levels than Europeans. Collectively, age, sex, and ethnicity accounted for 0.83% of the phenotypic variance in mtDNA-CN levels. All types of cell counts were significantly associated with mtDNA-CN including the standard set of covariates commonly adjusted for in mtDNA-CN investigations, namely, white blood cells (beta = −0.25 SDs in mtDNA-CN levels per 1 SD increase in cell counts; 95% CI, –0.25 to –0.24; $p < 2.23^{-308}$) and platelets (beta = 0.07; 95% CI, 0.06–0.07; $p < 2.23 \times 10^{-308}$) (*Figure 2B*). White blood cell and platelet counts each accounted for 6.3% and 0.5% of the variance in mtDNA-CN levels, respectively (*Figure 2B*; *Figure 2—source data 1*). Notably, neutrophil count (beta = −0.31; 95% CI, –0.31 to –0.30; $p < 2.23^{-308}$) was the strongest predictor of mtDNA-CN levels and accounted for more variance (9.9%) than white blood cell and platelet counts combined (8.2%). Collectively, total white blood cell, platelet, and neutrophil counts explained 12.3% variance in mtDNA-CN levels.

## GWAS identifies 70 common loci for mtDNA-CN

A GWAS was performed testing the association of 11,453,766 common genetic variants (MAF >0.005) with mtDNA-CN in 383,476 UKBiobank participants of European ancestry (*Figure 3*). In total, 9602 variants were associated with mtDNA-CN at genome-wide significance (*Figure 3A*; *Figure 3—figure*

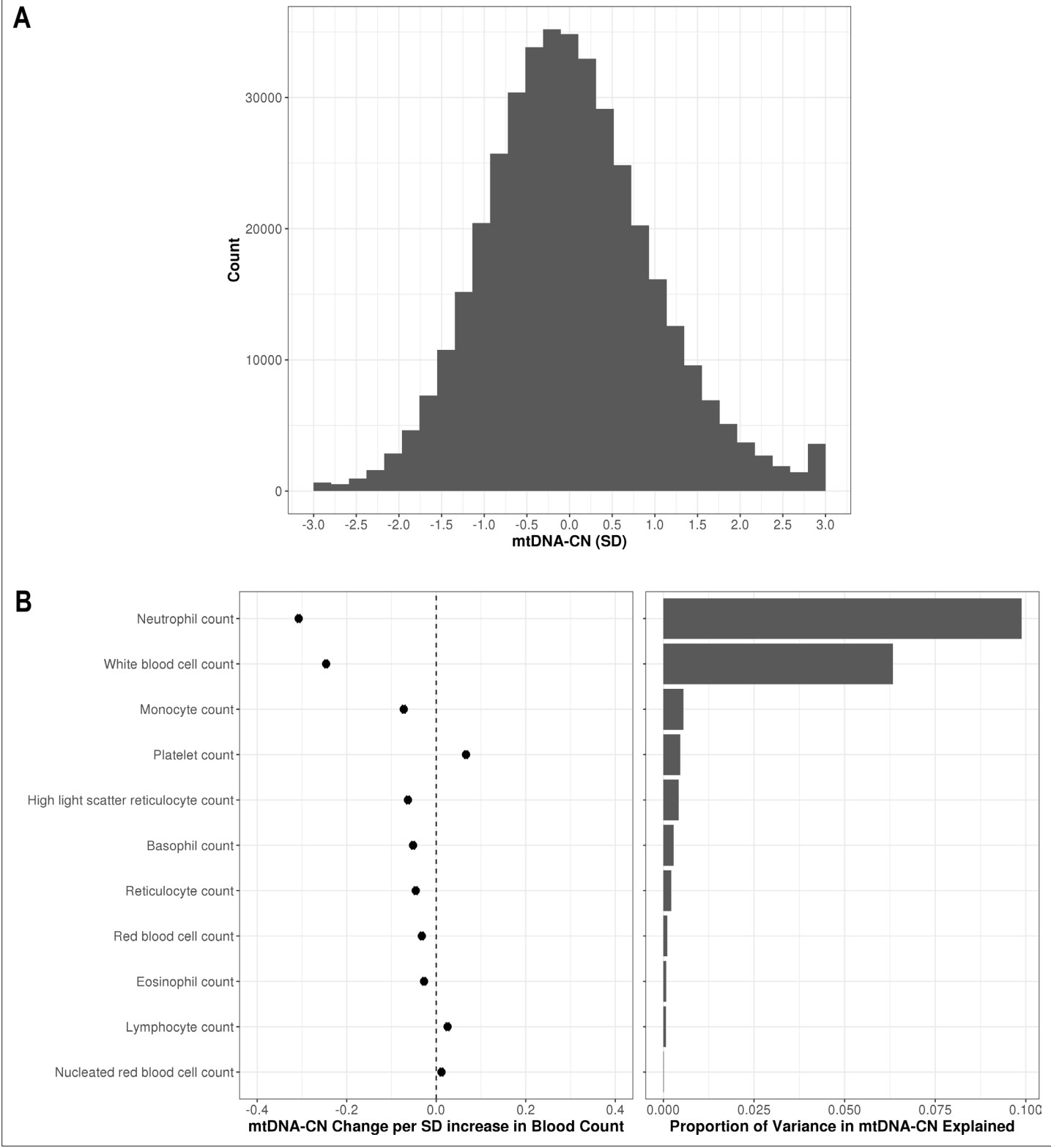

**Figure 2.** Distribution of automatic mitochondrial copy (AutoMitoC) estimates and the influence of blood cell counts. (**A**) Histogram illustrating AutoMitoC mitochondrial DNA copy number (mtDNA-CN) estimates in 395,781 UKBiobank participants expressed per SD change in mtDNA-CN. Associations between blood cell counts with mtDNA-CN levels as conveyed by forest plots illustrating effect estimates (left) and a bar plot showing the proportion of variance in mtDNA-CN explained (right). Models were adjusted for age, age², sex array type, 20 genetic principal components, and ethnicity. Both blood cell counts and mtDNA-CN levels were standardized (mean = 0; SD = 1).

*Figure 2 continued on next page*

*Figure 2 continued*

The online version of this article includes the following source data for figure 2:

**Source data 1.** Table showing the association of blood cell counts and AutoMitoC mtDNA-CN estimates in the UKBiobank.

*supplement 1*), encompassing 82 independent signals in 72 loci (*Supplementary file 2* – Tab 1; *Figure 3—source data 1*). The genomic inflation factor was 1.16 and the LD score intercept was 1.036, indicating that most inflation in test statistics was attributable to polygenicity. Sensitivity analyses revealed that nuclear mitochondrial DNA (NUMT) interference may have played a role in two independent signals (two loci), which were subsequently discarded, leading to a total of 80 independent signals in 70 loci. These 80 independent genetic signals explained 1.48% variance in mtDNA-CN levels.

Fine-mapping via the FINEMAP algorithm (*Benner et al., 2016*) yielded 95% credible sets containing 2363 genome-wide significant variants. Of the 80 independent genetic associations, 17 (22%) mapped to a single candidate causal variant; 32 (39%) mapped to 5 or fewer variants, and 42 (51%) mapped to 10 or fewer variants (*Figure 3B*; *Supplementary file 2* – Tab 2). Credible sets for 11 genetic signals overlapped with genes responsible for rare mtDNA depletion disorders including *DGUOK* (3), *MGME1* (2), *TFAM* (2), *TWNK* (2), *POLG2* (1), and *TYMP* (1) (*Supplementary file 2* – Tab 1 & 2). Several associations mapped to coding variants with high posterior probability. *DGUOK* associations mapped to a synonymous variant (rs62641680; posterior probability = 1) and a nonsynonymous variant (rs74874677; p=1). *TFAM* associations mapped to a 5' untranslated region (UTR) variant (rs12247015; p=1) falling within an ENCODE candidate cis-regulatory element with a promotor-like signature and an intronic variant (rs4397793; p=1) with a proximal enhancer-like signature. Last, *POLG2* associations mapped to a nonsynonymous variant (rs17850455; p=1). Beyond the six aforementioned mtDNA depletion genes identified at genome-wide significance, suggestive associations were found for *POLG* (rs2307441; $p=1.0 \times 10^{-7}$), *OPA1* (rs9872432; $p=5.2 \times 10^{-7}$), *SLC25A10* (rs62077224; $p=1.2 \times 10^{-7}$), and *RRM2B* (rs3907099; $p=4.7 \times 10^{-6}$). Given these observations, we hypothesized that mtDNA depletion genes may be generally enriched for common variant associations. Indeed, 10 (53%) of 19 known mtDNA depletion genes (*Oyston, 1998*) harbored at least suggestive mtDNA-CN associations ($p<5 \times 10^{-6}$).

We also compared our findings to genome-wide significant results from two recent GWAS of mtDNA-CN by *Hägg et al., 2021* and *Longchamps et al., 2022* to better contextualize our findings. Of the 66 independent signals identified by *Hägg et al., 2021*, we found varying levels of evidence to support 43 (65%) genetic associations; 41 (62%) variants were detected at Bonferroni significance; 2 (3%) variants were detected using a suggestive significance threshold ($p<5 \times 10^{-6}$) (*Supplementary file 2* – Tab 3). For the remaining 23 (35%) variants not detected at suggestive significance, we hypothesized that differences in covariate adjustment strategies. Specifically, Hagg et al. did not adjust for platelets which is an important confounder of blood mtDNA-CN levels which might explain the majority of these discrepancies. As a sensitivity analysis, we repeated association testing without adjusting for blood cell traits and were able to recover 19/23 (83%) of the remaining variants at suggestive significance threshold, which suggests that approximately 1 in 3 associations detected by *Hägg et al., 2021* may not be robust to adjustment for platelets (*Supplementary file 2* – Tab 3). While we could not perform the reverse lookup of our top GWAS hits within the *Hägg et al., 2021* GWAS due to a lack of full genome-wide summary statistics, 48 (60%) of our 80 genetic associations are likely novel relative to *Hägg et al., 2021* as these variants were not reported as genome-wide significant nor in linkage disequilibrium ($r^2 >0.10$) with their top hits (*Supplementary file 2* – Tab 1). Among the 133 independent signals identified by *Longchamps et al., 2022*, 119 achieved Bonferroni significance, of which 116 were available in our GWAS (*Supplementary file 2* – Tab 4). The remaining three variants (or their proxies; $r^2 >0.7$) did not meet the MAF threshold for inclusion (MAF >0.005). Of the 116 detectable variants, we found varying levels of evidence to support 112 (97%) genetic associations; 70 (60%) variants were detected at Bonferroni significance; 15 (13%) variants were detected using a suggestive significance threshold; 27 (24%) variants were detected at nominal significance with concordant directionality (*Supplementary file 2* – Tab 4). While we could not perform the reverse lookup of our top GWAS hits within the *Longchamps et al., 2022* GWAS, 11 (14%) of our 80 genetic associations are likely novel as these variants were not reported as genome-wide significant nor in linkage disequilibrium with their top hits (*Supplementary file 2* – Tab 1). Notably, these 11 associations were also not reported in *Hägg et al., 2021*.

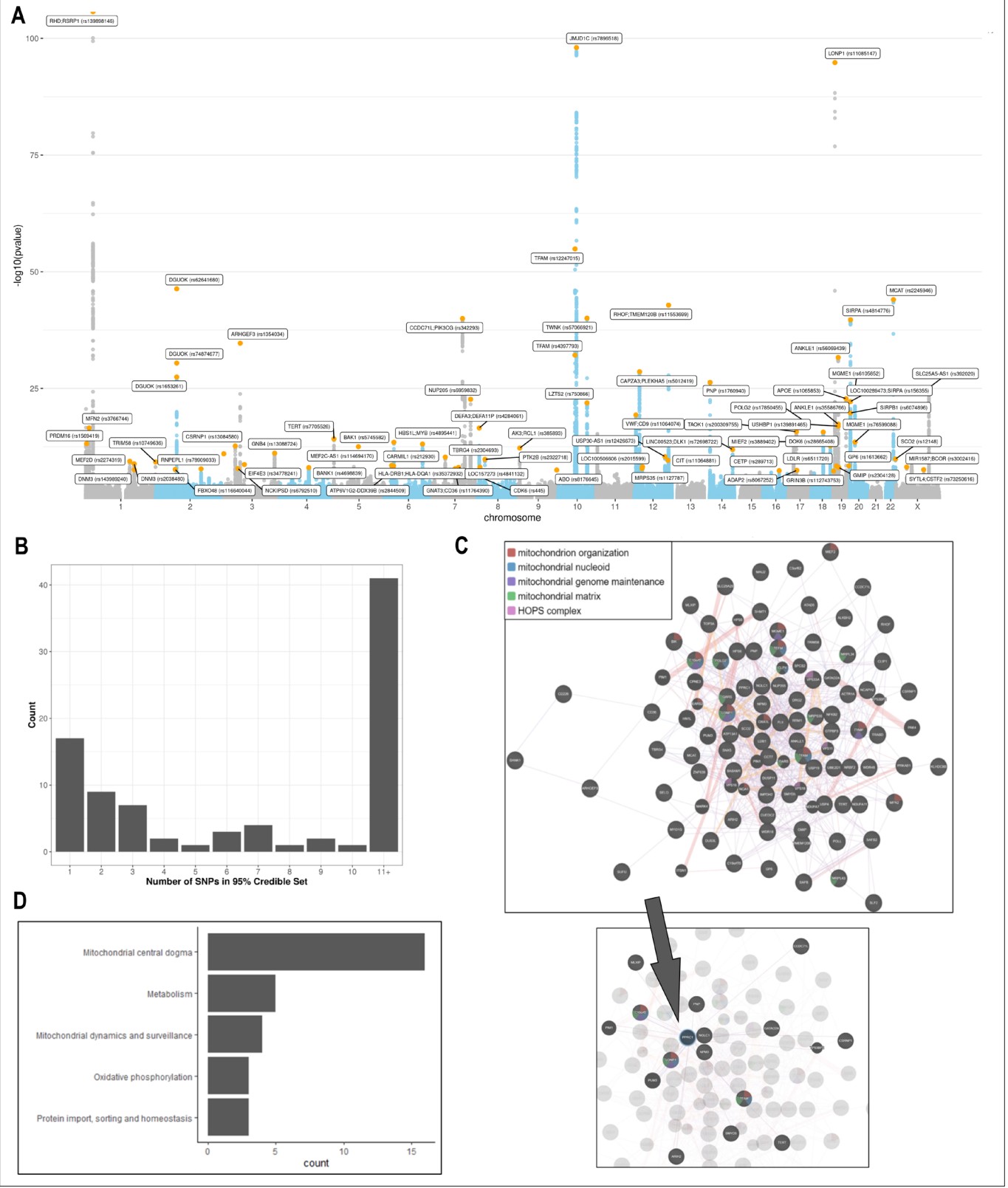

**Figure 3.** Analyses of common genetic loci associated with mitochondrial DNA copy number (mtDNA-CN). (**A**) Manhattan plot illustrating common genetic variant associations with mtDNA-CN. (**B**) Size distribution of 95% credible sets defined for 80 independent genetic signals. (**C**) GENE-MANIA-mania protein network interaction exploration showing all DEPICT and gene-mania prioritized genes (top) with functions color coded and a zoom-in

*Figure 3 continued on next page*

Figure 3 continued

highlighting interactors of the key mitochondrial (MT) regulator gene, *PPRC1* (bottom). (**D**) 'MitoPathway' counts corresponding to 27 prioritized MitoCarta3 genes encoding proteins with known MT localization.

The online version of this article includes the following source data and figure supplement(s) for figure 3:

**Source data 1.** Locus zoom plots for 72 loci and 82 conditionally independent genetic signals.

**Figure supplement 1.** Minor allele frequency (MAF) and ethnicity-stratified genome-wide association study quantile-quantile plots.

**Figure supplement 2.** Manhattan plot for transethnic genome-wide association study meta-analysis (N = 395,781).

**Figure supplement 3.** Correlation between conditionally independent mitochondrial DNA copy number loci effect estimates derived from European genome-wide association study (GWAS) meta-analyses (x-axes) vs effect estimates from non-European GWAS (y-axes).

Additionally, transethnic meta-analysis inclusive of non-Europeans (N = 395,781) was performed but given the small increase in sample size, GWAS findings remained highly similar (***Figure 3—figure supplement 2***). However, European effect estimates were significantly and highly correlated with those derived from South Asian ($r$ = 0.97; p=2.2 × 10$^{-15}$) and African ($r$ = 0.88; p=9.1 × 10$^{-5}$) GWAS analyses (***Figure 3—figure supplement 3***).

## mtDNA-CN loci influence MT gene expression and heteroplasmy

We postulated that differential expression of genes encoded by the MT genome may trigger changes in copy number, and thus a subset of identified loci may affect mtDNA-CN through MT genome transcription. *Ali et al., 2019* recently conducted a GWAS to identify nuclear genetic variants associated with variation in mtDNA-encoded gene expression (i.e. mitochondrial expression quantitative trait loci [mt-eQTLs]) (*Ali et al., 2019*). Nonsynonymous variants in *LONP1* (rs11085147) and *TBRG4* (rs2304693), as well as an intronic variant in *MRPS35* (rs1127787), were associated with changes in MT gene expression across various tissues (***Supplementary file 2*** – Tab 5). Nominally associated mtDNA-CN loci were also observed to influence MT gene expression including intronic variants in both *PNPT1* (rs62165226; mtDNA-CN p=5.5 × 10$^{-5}$) and *LRPPRC* (rs10205130; mtDNA-CN p=1.1 × 10$^{-4}$). Although differences in MT gene expression may be a consequence rather than a cause of variable mtDNA-CN, the analysis performed by *Ali et al., 2019* was corrected for factors associated with global changes in the MT transcriptome (*Ali et al., 2019*). Moreover, the direction of effect estimates between mtDNA-CN and mt-eQTLs varied depending on gene and tissue context. Altogether, such findings imply that some mtDNA-CN loci may regulate mtDNA-CN by influencing MT gene expression.

Heteroplasmy refers to the coexistence of multiple mtDNA alleles within an individual for a particular variant, which is a function of the multicopy nature of the MT genome. A recent GWAS by *Nandakumar et al., 2021* for mean heteroplasmy levels in saliva specimen provided initial evidence supporting a shared genetic basis for heteroplasmy and copy number. To further explore the overlap in genetic determinants of these traits, we searched for the previously reported heteroplasmy loci within our mtDNA-CN GWAS. Of 19 matching variants between the GWAS, 4 heteroplasmy loci were also associated with mtDNA-CN at genome-wide significance including variants nearby or within *TINCR/LONP1* (rs12461806; mtDNA-CN GWAS p=7.5 × 10$^{-88}$), *TWNK/MPRL43* (rs58678340; p=1.3 × 10$^{-39}$), *TFAM* (rs1049432; p=1.5 × 10$^{-21}$), and *PRKAB1* (rs11064881; p=2.6 × 10$^{-10}$) genes. Consistent with the initial finding from *Nandakumar et al., 2021* that the heteroplasmy-increasing *TFAM* allele was also associated with higher mtDNA-CN, we also observed concordant directionality for the other three variants. No additional mtDNA heteroplasmy loci were identified to influence mtDNA-CN when using a more liberal suggestive significance threshold. Full genome-wide summary statistics for the heteroplasmy GWAS were not publicly available so we could not perform the reverse lookup to examine whether the 80 mtDNA-CN variants affected heteroplasmy.

## Genes and pathways implicated in the regulation of mtDNA-CN

DEPICT analysis led to the prioritization of 91 out of 18,922 genes (false discovery rate (FDR) p<0.05; ***Supplementary file 2*** – Tab 6). Among them, 87 of the genes intersected with the GeneMANIA database and were uploaded to the GeneMANIA platform to identify additional functionally related genes (*Warde-Farley et al., 2010*). GeneMANIA analysis discovered an additional 20 related genes (***Supplementary file 2*** – Tab 7). Among the 107 total genes prioritized by DEPICT or GeneMANIA

(*Figure 3C*), MT functions were significantly enriched in gene ontology (GO) terms including mitochondrion organization (coverage: 12/225 genes; FDR p=7.4 × 10⁻⁵), MT nucleoid (6/34; FDR p=2.2 × 10⁻⁴), MT genome maintenance (4/10; FDR p=6.8 × 10⁻⁴), and MT matrix (11/257; FDR p=6.8 × 10⁻⁴). Visual inspection of the links between key genes involved in these functions highlights *PPRC1*, a member of the Peroxisome proliferator-activated receptor-gamma coactivator-1alpha (PGC-1A) family of MT biogenesis activators (*Scarpulla, 2011*), as a potential coordinator of mtDNA-related processes (*Figure 3C*).

MitoCarta3 is a comprehensive and curated inventory of 1136 human proteins (1120 nuclear) known to localize to the mitochondria based on experiments of isolated mitochondria from 14 nonblood tissues (*Rath et al., 2021*). We leveraged this recently updated database, that was absent from GeneMANIA, to conduct a complementary set of targeted analyses focused on MT annotations (*Supplementary file 2* – Tab 7). First, we hypothesized that prioritized genes would be generally enriched for genes encoding the MT proteome. Overall, 27 (25%) of 107 genes had evidence of MT localization corresponding to a 4.2-fold enrichment (null expectation = 5.9%; p=1.0 × 10⁻¹⁰). Next, given that *PPRC1*, an activator of MT biogenesis, was prioritized by DEPICT analyses and then linked to central mtDNA regulators in GeneMANIA, we postulated that prioritized genes may be enriched for downstream targets of PGC-1A. PGC-1A induction resulted in a higher mean fold change among prioritized genes (beta = 1.48; 95% CI, 0.60–2.37) as compared to any MT gene (beta = 1.19; 95% CI, –0.76–3.13; t-test p=0.04). Finally, we categorized the 27 MitoCarta3 genes into their respective pathways. Most (16; 57%) genes were members of the 'Mitochondrial central dogma' pathway, which represents a nearly threefold enrichment as compared to the frequency of this pathway in the whole MitoCarta3 database (null expectation = 20.7%; p=1.3 × 10⁻⁵). Other implicated (albeit not significantly enriched) pathways included 'Metabolism,' 'Mitochondrial dynamics and surveillance,' 'Oxidative phosphorylation,' and 'Protein import, sorting and homeostasis' (*Figure 3D*). Four proteins were annotated as part of multiple pathways including *TYMP/SCO2, GTPBP3, MIEF1,* and *OXA1L* (*Supplementary file 2* – Tab 7).

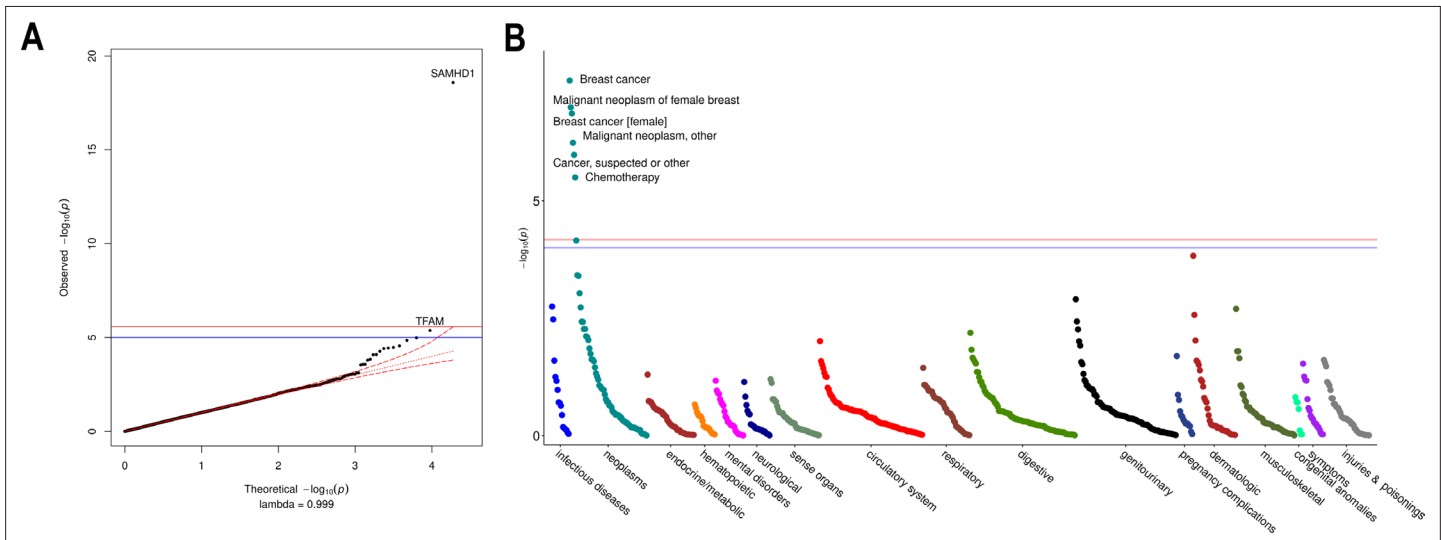

**Figure 4.** Rare variant gene burden association testing with mitochondrial DNA copy number and disease risk. (**A**) QQ plot illustrating expected vs observed -log10 p-values for exome-wide burden of rare (MAF <0.001) and nonsynonymous mutations. (**B**) Manhattan plot showing phenome-wide significant associations between *SAMHD1* carrier status and cancer-related phenotypes.

The online version of this article includes the following source data and figure supplement(s) for figure 4:

**Source data 1.** Summary statistics for exome-wide association testing of rare protein-altering variants with mtDNA-CN.

**Source data 2.** Summary statistics for phenome-wide association testing of SAMHD1 carrier status.

**Figure supplement 1.** Effect of rare mutation carrier status for *SAMHD1* and *TFAM* genes on mitochondrial DNA copy number (mtDNA-CN) levels.

# Exome-wide association testing uncovers rare coding *SAMHD1* mutations as a determinant of mtDNA-CN levels and breast cancer risk

We performed an ExWAS in 147,740 UK Biobank participants with WES data to assess the contribution of rare coding variants. Among 18,890 genes tested, *SAMHD1* was the only gene reaching exome-wide significance (*Figure 4A*; *Figure 4—source data 1*). The carrier prevalence of rare *SAMHD1* mutations was 0.75%, and on average, mutation carriers had higher mtDNA-CN than noncarriers (beta = 0.23 SDs; 95% CI, 0.18–0.29; $p=2.6 \times 10^{-19}$; *Figure 4—figure supplement 1*). Also, while none of the 19 known mtDNA depletion genes reached Bonferroni significance, a suggestive association was found for *TFAM* (beta = −0.33; 95% CI, −0.47 to −0.19; $p=4.2 \times 10^{-6}$), and this association was independent of the common *TFAM* variants (rs12247015; rs4397793) previously identified in the GWAS (beta = −0.33; 95% CI, −0.47 to −0.19; $p=8 \times 10^{-6}$). Rare variants in *SAMHD1* and *TFAM* accounted for 0.06% of the variance in mtDNA-CN levels. Collectively, rare and common loci accounted for 1.55%.

To evaluate whether rare *SAMHD1* mutations also influenced disease risk, we conducted phenome-wide association testing of 771 diseases within the UKBiobank. At phenome-wide significance, *SAMHD1* mutation carrier status was associated with approximately twofold increased risk of breast cancer (OR = 1.91; 95% CI, 1.52–2.40; $p=2.7 \times 10^{-8}$), as well as greater risk of 'cancer (suspected or other)' (OR = 1.52; 95% CI, 1.28–1.80; $p=1.1 \times 10^{-6}$; *Figure 4B*; *Figure 4—source data 2*). Exclusion of breast cancer cases was attenuated but did not nullify the association with 'cancer (suspected or other)' (OR = 1.36; 95% CI, 1.10–1.67; p=0.004) suggesting that *SAMHD1* mutations may also increase risk of other cancers, as has been shown for colon cancer (*Rentoft, 2019*). To understand whether differences in mtDNA-CN levels between *SAMHD1* mutation carriers was a consequence of cancer diagnosis, we repeated association testing with mtDNA-CN excluding cancer patients. In this

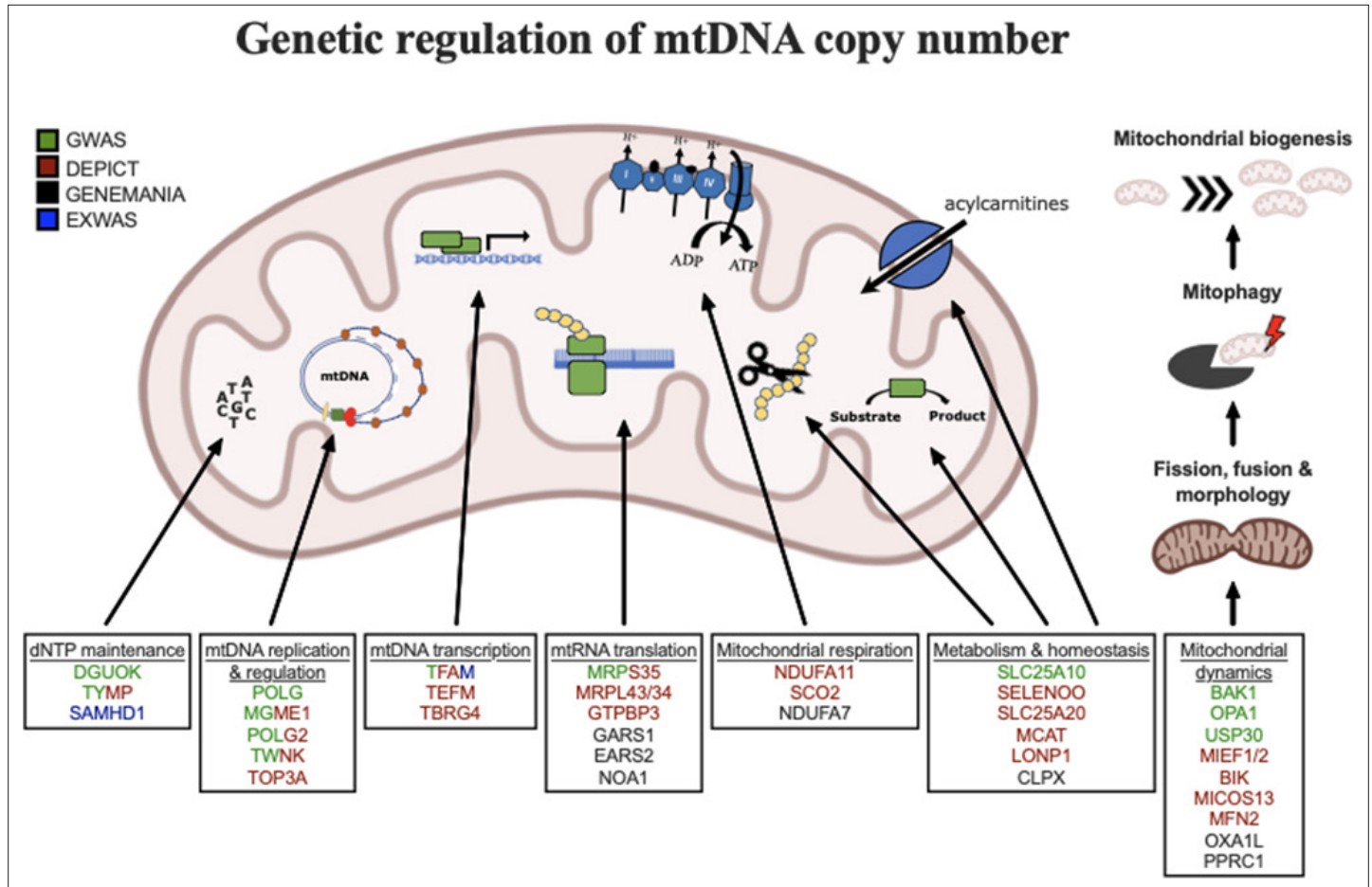

**Figure 5.** Graphical summary of mitochondrial genes and pathways implicated by genetic analyses. Color coding indicates through which set(s) of analyses genes were identified. The image was generated using BioRender (https://biorender.com/).

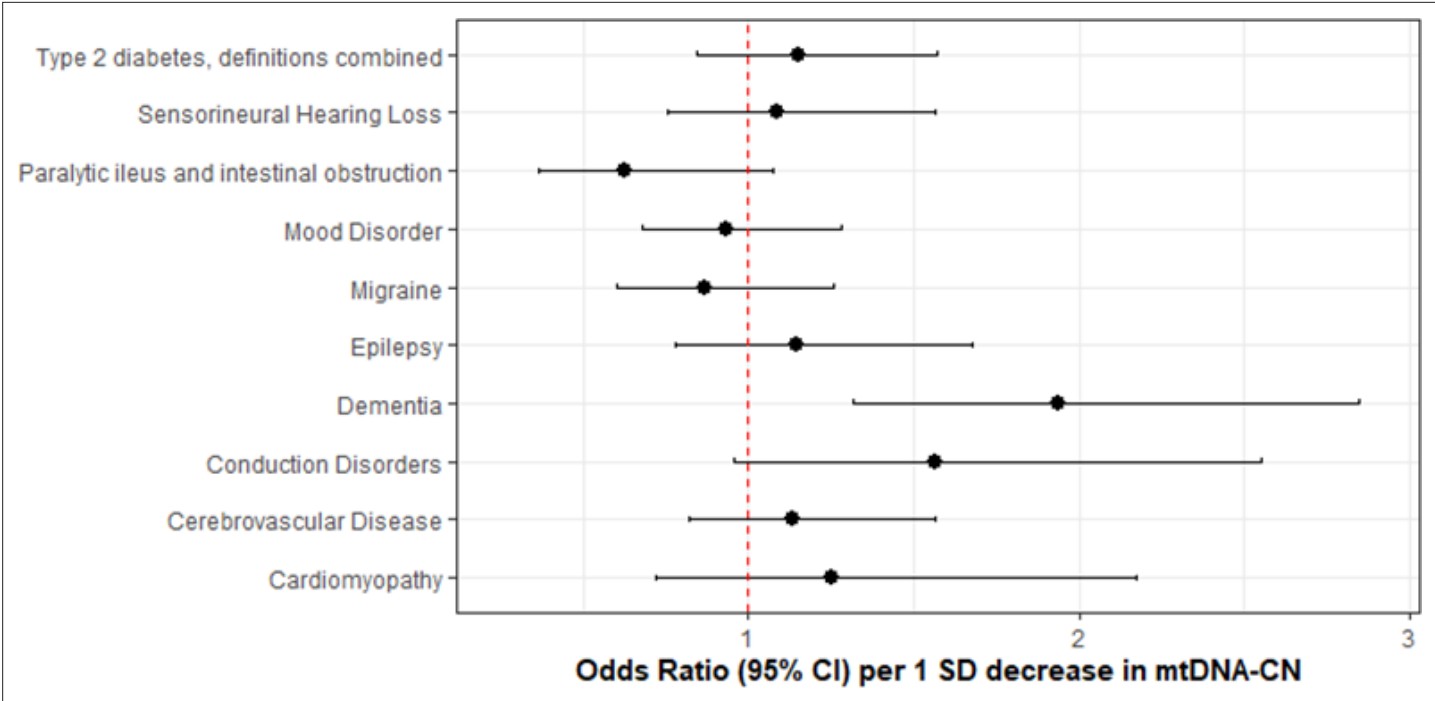

**Figure 6.** Association of genetically lower mitochondrial DNA copy number with mitochondrial (MT) disease phenotypes. Coefficient plots for Mendelian randomization analyses of MT disease traits. In the absence of heterogeneity (Egger intercept p≥0.05; MR-PRESSO global heterogeneity p≥0.05), the inverse variance weighted result was reported. In the presence of balanced pleiotropy (MR-PRESSO global heterogeneity p<0.05), the weighted median result was reported. No set of analyses had evidence for directional pleiotropy (Egger intercept p<0.05).

The online version of this article includes the following source data for figure 6:

**Source data 1.** Mendelian randomization analyses of mtDNA-CN versus mitochondrial disease phenotypes.

analysis, the association with mtDNA-CN levels was not attenuated (beta = 0.26; 95% CI, 0.19–0.32; p=7.8 × 10$^{-15}$) suggesting that the effect of rare *SAMHD1* variants on mtDNA-CN levels is not driven by its relationship with cancer status. A summary of MT genes and pathways implicated by common and rare loci is provided in *Figure 5*.

## Mendelian randomization analysis implicates low mtDNA-CN as a causal mediator of dementia

Given that common variant loci overlapped with several mtDNA depletion genes, we postulated that polygenically low mtDNA-CN might cause a milder syndrome with phenotypically similar manifestations. To assess whether mtDNA-CN may represent a putative mediator of mtDNA depletion-related phenotypes, we conducted Mendelian randomization analyses between genetically determined mtDNA-CN and MT disease phenotypes using summary statistics derived from the FinnGen v4 GWAS dataset (*Supplementary file 2* – Tab 8).

After accounting for multiple testing of 10 phenotypes, an association between mtDNA-CN and all-cause dementia was found (OR = 1.94 per 1 SD decrease in mtDNA-CN; 95% CI, 1.55–2.32; p=7.5 × 10$^{-4}$; *Figure 6*; *Figure 6—source data 1*). Sensitivity analyses indicated no evidence of global (MR-PRESSO p=0.51; Q-statistic p=0.51) or directional (Egger Intercept p=0.47) pleiotropy. The 27 selected variants accounted for 0.70% of the variance in mtDNA-CN and 0.13% of the risk for dementia, consistent with a causal effect of mtDNA-CN on dementia risk and not vice versa (Steiger p=1.9 × 10$^{-62}$). Findings were robust across several different MR methods including the weighted median (OR = 2.47; 95% CI, 1.93–3.00; p=0.001), MR-EGGER (OR = 2.41; 95% CI, 1.71–3.11; p=0.02), and GSMR-HEIDI (OR = 1.95; 95% CI, 1.34–2.84; p=0.001) methods. Results also remained statistically significant when using a broader set of genetic instruments including all genome-wide significant loci irrespective of whether genetic variants were located proximally to MitoCarta3 genes (GSMR-HEIDI OR = 1.31; 95% CI, 1.02–1.68; p=0.04). Furthermore, we replicated the dementia association using a

second UKBiobank-independent GWAS dataset derived from the International Genomics of Alzheimer's Disease Consortium (2013) including 17,008 Alzheimer's disease patients (OR = 1.41; 95% CI, 1.0001–1.98; p=0.04993) (*Lambert et al., 2013*).

## Discussion

We developed a novel method to estimate mtDNA-CN from genetic array data, 'AutoMitoC,' and applied it to the UKBiobank study. Extensive genetic investigations led to several key insights regarding mtDNA-CN. First, several novel common and rare genetic determinants of mtDNA-CN were identified, totaling 71 loci. Second, these loci were enriched for MT processes related to dNTP metabolism and the replication, packaging, and maintenance of mtDNA. Third, we observed a strong role for common variation within known mtDNA depletion genes in regulating mtDNA-CN in the general population. Fourth, we found that rare variants in *SAMHD1* not only affect mtDNA-CN levels but also confer risk to cancer. Finally, we provided the first Mendelian randomization evidence implicating low mtDNA-CN as a causative risk factor for dementia.

While several investigations for mtDNA-CN have been published, the present study represents the most comprehensive genetic assessment published to date (*Cai et al., 2015*; *Longchamps, 2019*; *Hägg et al., 2021*). Notably, *Hägg et al., 2021* recently conducted the largest GWAS for mtDNA-CN in 295,150 UKBiobank participants and identified 50 common loci (*Hägg et al., 2021*). However, the method developed by *Hägg et al., 2021* calibrated SNP probe intensities based on association with WES read depths, which may limit the convenience of the method. In contrast, AutoMitoC only necessitates array probe intensities and does not require any secondary genetic measurements (WES or otherwise) for calibration. In addition, AutoMitoC exhibits superior concordance with WES-based estimates (Hagg r = 0.33; AutoMitoC r = 0.45), which were validated in an independent dataset with gold standard qPCR measurements (AutoMitoC r = 0.64). Further, *Hägg et al., 2021* restricted genetic analyses to unrelated European individuals, whereas we incorporated ~100,000 additional individuals and demonstrated consistency in genetic effects between Europeans and non-Europeans ($r \geq 0.88$). The greater sample size in combination with more accurate mtDNA-CN estimates may explain the additional 48 (59%) independent common genetic associations identified at genome-wide significance. Furthermore, we took additional precaution to ensure that cataloged NUMTs were not influencing our genetic association results and also to empirically evaluate the robustness of each genetic association against cataloged and uncataloged NUMTs. Finally, in the present study we included complementary explorations of the role of rare variants through ExWAS and phenome-wide association study (PheWAS), as well as, Mendelian randomization analyses to assess disease contexts whereby mtDNA-CN may represent a causal mediator and a potential therapeutic target.

MtDNA-CN has proven to be a biomarker of cardiovascular disease in several large epidemiological studies, with studies often assuming that such relationships are attributable to pathological processes including MT dysfunction, oxidative stress, and inflammation (*Tin et al., 2016*; *Wu et al., 2017*; *Fazzini et al., 2019*; *Koller et al., 2020*). Consistent with previous GWAS findings, our genetic analyses confirm that mtDNA-CN indeed reflects specific MT functions, but perhaps not the ones commonly attributable to mtDNA-CN (*Cai et al., 2015*; *Guyatt et al., 2019*; *Hägg et al., 2021*). Primarily, differences in mtDNA-CN reflect MT processes related to dNTP metabolism and the replication, maintenance, and organization of mtDNA. Secondarily, genes involved in MT biogenesis, metabolism, oxidative phosphorylation, and protein homeostasis were also identified but do not represent the main constituents. The observed enrichment in common variant associations within mtDNA depletion genes further reinforces the notion that differences in mtDNA-CN first and foremost reflect perturbations in mtDNA-related processes.

No therapy for mtDNA depletion disorders currently exists with treatment mainly consisting of supportive care. Intriguingly, we found that rare variants within *SAMHD1* were associated with increased levels of mtDNA-CN. *SAMHD1* is a multifaceted enzyme with various functions including tumor suppression through DNA repair activity and maintenance of steady-state intracellular dNTP levels, which has been involved in HIV-1 replication (*Baldauf et al., 2012*; *Kretschmer et al., 2015*). Rare homozygous and compound heterozygous loss-of-function mutations in *SAMHD1* result in an immune encephalopathy known as Aicardi Goutieres' syndrome (*White et al., 2017*). Imbalanced intracellular dNTP pools and chronic DNA damage cause persistent elevations in interferon alpha thus mimicking a prolonged response to HIV-1 infection. While Aicardi Goutieres' syndrome is a severe

recessive genetic disorder, case reports of *SAMHD1*-related disease often describe heterozygous parents and siblings as being unaffected or with milder disease (familial chilblain lupus 2) (*Haskell et al., 2018*). In the UKBiobank, the vast majority (99.4%) of individuals possessing *SAMHD1* mutations were heterozygote carriers, who had a twofold increased risk of breast cancer. Indeed, our finding that *SAMHD1* mutations associate with both elevated mtDNA-CN levels and risk of breast cancer belies the prevailing notion that higher mtDNA-CN is always a protective signature of proper MT function and healthy cells. Such findings may have important clinical implications for genetic screening. First, heterozygous *SAMHD1* mutations may be an overlooked risk factor for breast cancer considering that ~1 in 130 UKBiobank participants possessed a genetic mutation conferring twofold elevated risk. Notably, while *SAMHD1* mutations have been described previously to be associated with various cancers (*Kohnken et al., 2015*; *Rentoft, 2019*), this gene is not routinely screened nor part of targeted gene panels outside the context of neurological disorders (https://www.genedx.com/test-catalog/available-tests/comprehensive-common-cancer-panel/). Second, unaffected parents and siblings of Aicardi Goutieres patients might also present with greater risk of cancer. Third, while *SAMHD1* is a highly pleiotropic protein, therapeutic strategies to dampen (but not abolish) *SAMHD1* activity might be considered to treat mtDNA depletion disorders caused by defects in nucleotide metabolism. Indeed, *Franzolin et al., 2015* demonstrated that siRNA knockdown of *SAMHD1* in human fibroblasts with *DGUOK* mtDNA depletion mutations partially recovered mtDNA-CN (*Franzolin et al., 2015*).

To our knowledge, we provide the first Mendelian randomization evidence that mtDNA-CN may be causally related to risk of dementia. Although dysfunctional mitochondria have long been implicated in the pathogenesis of Alzheimer's disease, only recently has mtDNA-CN been tested as a biomarker. *Silzer et al., 2019* conducted a matched case-control study of 46 participants and showed that individuals with cognitive impairment had significantly lower blood-based mtDNA-CN (*Silzer et al., 2019*). *Andrews and Goate, 2020* studied the relationship between postmortem brain tissue mtDNA-CN and measures of cognitive impairment in 1025 samples (*Andrews and Goate, 2020*). Consistent with our findings, a 1 SD decrease in brain mtDNA-CN was associated with lower mini mental state exam (beta = –4.02; 95% CI, –5.49 to –2.55; p=$1.07 \times 10^{-7}$) and higher clinical dementia rating (beta = 0.71; 95% CI, 0.51–0.91). Both studies implicate blood- and brain-based mtDNA-CN as a marker of dementia but were retrospective. In contrast, *Yang et al., 2021* observed a significant association between mtDNA-CN and incident risk of neurodegenerative disease (Parkinson's and Alzheimer's disease)(*Yang et al., 2021*). Altogether, our results combined with previous findings suggest that mtDNA-CN represents both a marker and mediator of dementia. Considering that our overall findings suggest that mtDNA-CN reflects numerous MT subprocesses, future studies will be required to disentangle which ones, as reflected by diminished mtDNA-CN, truly mediate dementia pathogenesis.

Several limitations should be noted. First, mtDNA-CN approximated by array-based methods remains imperfectly accurate as compared to qPCR or whole genome sequencing measurements, though we found strong correlation between AutoMitoC and qPCR-based estimates in this study (r = 0.64; p<$2.23 \times 10^{-308}$). AutoMitoC was benchmarked using multiple comparative mtDNA-CN estimation modalities, in independent cohorts, and in various ethnic groups, but further work can be done to assess the reproducibility and error of mtDNA-CN estimates, calibration to absolute counts, and the robustness to different genotyping platforms. Although whole genome sequencing will eventually supplant array-based mtDNA-CN GWAS, we hypothesize that the improvements made in the areas of speed, portability to ethnically diverse studies, and ease-of-implementation, should greatly increase accessibility of mtDNA-CN research as a plethora of genotyping array data is presently available to reanalyze. Second, we did not detect strong evidence of confounding from population structure (or substructure) in our genetic analyses, and we undertook several precautions to mitigate such biases including (i) stratified GWAS analyses by ethnicity, (ii) adjustment for a large number of intraethnic principal components (PCs), and (iii) restriction of downstream analyses to use of European GWAS summary statistics to harmonize with external databases; however, it is still plausible that population substructure may have influenced our findings. Third, Mendelian randomization analyses were underpowered to conduct a broad survey of diseases in which MT dysfunction may play a causal role, and even among the 10 phenotypes tested, statistical power varied by disease prevalence as evidenced by relatively large CI for the dementia association. Accordingly, future studies including larger case

sample sizes are necessary to provide more precise effect estimates. Equally as important, we were unable to differentiate whether specific MT subpathways mediated risk of disease. As additional loci are uncovered, such analyses may be feasible. Fourth, variants and genes implicated in the regulation of mtDNA-CN may be specific to blood samples though findings suggest that many mtDNA-CN loci act through genes that are widely expressed in mitochondria across multiple tissues. Future studies are required to determine whether associations are ubiquitous across mitochondria-containing cells and to investigate the role of mtDNA-CN in other tissues. Lastly, whole blood mtDNA-CN reflects a heterogenous mixture of nucleated and unnucleated cells, and despite adjustment for major known confounding cell types, interindividual differences in cell subpopulations not captured by a standard blood cell count may represent an important source of confounding.

## Conclusion

Although commonly viewed as a simple surrogate marker for the number of mitochondria present within a sample, genetic analyses suggest that mtDNA-CN is a highly complex biomarker under substantial nuclear genetic regulation. mtDNA-CN reflects a mixture of MT processes mostly pertaining to mtDNA regulation. Accordingly, the true relationship between mtDNA-CN measured in blood samples with human disease remains to be completely defined though we find evidence for mtDNA-CN as a putative causal risk factor for dementia. Future studies are necessary to decipher if mtDNA-CN is directly involved in the pathogenesis of dementia and other diseases or whether other specific MT processes are truly causative.

# Materials and methods
## The AutoMitoC number pipeline
### Background and rationale

While SNP array data is intended for highly multiplexed determination of genotypes, the raw probe signal intensities used for genotypic inference can also be co-opted to derive estimates of mtDNA-CN. Determining genotypes for a sample at a given variant site relies on contrasting hybridization intensities of allele-specific oligonucleotide probes and then assigning membership to the most probable genotype cluster based on intensity properties (*Figure 1—figure supplement 1*). Variation in intensities within each genotyping cluster can also be co-opted to deduce variations in copy number. A commonly used metric of probe intensity is the 'L2R,' which denotes $\log_2$(observed intensity/expected intensity), where the expected intensity is defined as the median signal intensity for a probe conditional on each genotype cluster.

The 'MitoPipeline' is a framework for estimating mtDNA-CN from array-based L2R (aka L2R) values developed by *Lane, 2014* (*Lane, 2014*; *Zhang et al., 2017*). An overview of the MitoPipeline is described in *Figure 1—figure supplement 2*. To briefly summarize: (i) autosomal and MT L2R values are first corrected for GC waves; (ii) a high-quality set of MT and autosomal markers are selected largely based on visual inspection of genotype clusters and basic local alignment search tool (BLAST) alignment for nonhomologous sequences; (iii) PCA of at least 40,000 autosomal markers is conducted to capture batch effects; (iv) finally, mtDNA-CN is estimated based on median MT L2R value for each sample and then corrected for background noise through residualization of top autosomal PCs.

The MitoPipeline has proven to be effective in estimating mtDNA-CN (correlation coefficient *R* ~0.5 with direct qPCR estimates) as evidenced by multiple epidemiological studies employing this method (*Ashar et al., 2017*; *Zhang et al., 2017*; *Fazzini et al., 2019*). First, visual inspection of MT probe intensity clusters is recommended to remove probes with poorly differentiated genotype clusters. However, this process is time-consuming (especially for biobank studies that are genotyped across thousands of batches); determination of probes with 'good' vs 'bad' genotype clustering is subjective; and guidance is only provided for adjudication of polymorphic but not rare or monomorphic markers which may still be informative. Second, in consideration of nuclear and MT sequences with significant sequence similarity due to past and recurrent transposition of MT sequence into the nuclear genome, also known as 'NUMTs' (*Simone et al., 2011*), the MitoPipeline recommends exclusion of MT probes with greater than 80% sequence similarity to the nuclear genome. While determining sequence homology of probes to the nuclear genome may have been feasible with older microarrays wherein probe sequences were often publicized, for many contemporary arrays, including

the UKBiobank array, such information is not readily available. Third, LD pruning of common auto-somal variants is required to ascertain a set of independent genetic variants, but implementation of this approach within ethnically diverse studies becomes more complex since genetic independence is ancestry dependent. Under the MitoPipeline framework, each ethnicity warrants a unique set of common variants, which not only adds to computational burden but also creates an additional source of variability in terms of performance of the method between ethnicities.

Therefore, we developed a new array-based mtDNA-CN estimation method, which we have dubbed the 'AutoMitoC' pipeline, which incorporates three key amendments: (i) autosomal signal normalization utilizes globally rare variants in place of common variants which confer advantages in terms of both speed and portability to ethnically diverse studies, (ii) cross-hybridizing probes are identified by assessing evidence for cross-hybridization via association of signal intensities (rather than using genotype association and identification of homologous sequences through BLAST alignment), and (iii) the primary estimate of MT signal is ascertained using PCA as opposed to using the median signal intensity of MT probes. The rationale underlying these amendments are described in detail in the subsequent sections. An overview of the AutoMitoC pipeline is provided in *Figure 1*.

## Development of AutoMitoC in the UKBiobank study

To develop the AutoMitoC pipeline we used genetic datasets from the UKBiobank. The UKBiobank is a prospective cohort study including approximately 500,000 UK residents (ages 40–69 years) recruited from 2006 to 2010 in whom extensive genetic and phenotypic investigations have been and continue to be done (*Sudlow et al., 2015*). All UKBiobank data reported in this manuscript were accessed through the UKBiobank data showcase under application #15,525. All analyses involve the use of genetic and/or phenotypic data from consenting UKBiobank participants.

Two main genetic datasets from the UKBiobank were incorporated in the development of AutoMitoC. First, CNV L2R values derived from genetic arrays (i.e. normalized array probe intensities) for 488,264 samples were downloaded using the *ukbgene* utility, and their corresponding genotype calls (data field: 22418) were downloaded with *gfetch*. Second, exome alignment maps (EXOME FE CRAM files and indices; data fields 23163 & 23164) from the first tranche of 49,989 samples released in March, 2019 were downloaded with the 'ukbfetch' utility. L2R values were used to derive AutoMitoC mtDNA-CN estimates. For WES data, samtools *idxstats* was used to derive the number of sequence reads aligning to MT and autosomal genomes, from which an estimate of mtDNA-CN was derived according to the procedure by *Longchamps, 2019*; these complementary WES-based mtDNA-CN estimates served as a comparator to benchmark AutoMitoC performance.

Initial quality control of 488,264 samples and 784,256 directly genotyped variants was executed in PLINK followed that recommended by the MitoPipeline (i.e. sample call rate >0.96; variant call rate >0.98; Hardy Weinberg equilibrium [HWE] p>1 × 10⁻⁵; PLINK mishap p>1 × 10⁻⁴; genotype asso-ciation with sex p>0.00001; LD pruning r2 <0.30; MAF >0.01) (*Purcell et al., 2007*). Variants within 1 Mb of immunoglobulin, T-cell receptor genes, and centromeric regions were removed. After this quality control procedure, 466,093 samples and 86,677 common variants remained. Next, genomic waves were corrected according to *Diskin et al., 2008* using the PennCNV 'genomic_wave.pl' script (https://github.com/WGLab/PennCNV/blob/master/genomic_wave.pl) (*Wang et al., 2007*; *Diskin et al., 2008*). Samples with high genomic waviness (L2R SD >0.35) before and after GC correction were removed resulting in 431,501 samples with array L2R values corresponding to 86,677 common autosomal variants. Last, we excluded samples representing blood cell count outliers ( >3 SDs) as per *Longchamps, 2019* which led to 395,781 participants (*Longchamps, 2019*). Finally, we took the intersection of European samples with both suitable array and WES data resulting in a final testing dataset of 34,436 European participants. To evaluate the possibility of replacing common autosomal variant signal normalization with rare variants, we also analyzed a set of 79,611 rare variants with an MAF <0.01.

## Background correlation between autosomal and MT L2R values

Inference of relative mtDNA-CN from array data consists of determining the ratio of MT to auto-somal probe signal intensities (or normalized L2R values) within each sample. Technical (e.g. batch and plate effects) and latent sample factors confound raw signal intensities for reasons unrelated to DNA

quantity. Such confounders induce strong cross-genome correlation, and therefore, a necessary first step is to remove this background noise from autosomal and MT probe intensities.

Indeed, even after correcting autosomal L2R values for genomic waves, we observed significant correlation between individual autosomal probe intensities and the median sample intensity across the 265 MT variants (*Figure 1—figure supplement 3A*). The extent of cross-genome intensity correlation varied based on MAF, with rare autosomal variants (MAF <0.01; M = 79,611) showing the strongest correlation. We postulate that intensity properties for rare variants, which have a higher prevalence of homozygous genotypes, more strongly resemble those for MT genotypes, which are predominantly homoplasmic. On this basis, we only use rare autosomal variants to represent autosomal signal. While this approach contrasts with the MitoPipeline, which utilizes common genetic variants to represent autosomal signal, restricting autosomal signal normalization to rare variants confers three major advantages while maintaining the same level of concordance with WES estimates ($r_{common}$ = 0.50; $r_{rare}$ = 0.49). First, this allows for further streamlining of the pipeline as this precludes the necessity for common variant filters, such as LD pruning. Second, we show that fewer PCs are necessary to capture the same proportion of total variance in signal intensities with rare as opposed to common variants. Approximately 70% of the total variance in rare autosomal intensities was explained by 120 PCs, whereas the same proportion of variance in common autosomal intensities would necessitate more than 1000 PCs (*Figure 1—figure supplement 3B*). In the UKBiobank, PCs were derived via the eigendecomposition of the empirical covariance matrix conducted in Python 3.6, using NumPy and SciPy (*Harris et al., 2020*; *Virtanen et al., 2020*). Third, the set of autosomal markers used in deriving mtDNA-CN remains independent from the set of common autosomal variants analyzed in subsequent GWAS for mtDNA-CN. Effectively, this ensures that common autosomal variants evaluated for association with mtDNA-CN in downstream GWAS analyses are not directly incorporated into autosomal signal normalization, which could otherwise attenuate GWAS signals.

## Empirical detection of off-target probes

After adjustment for 120 autosomal PCs (approximating the elbow of the variance explained curve), there persisted a smaller subset of autosomal probes that were significantly correlated with median MT intensity (*Figure 1—figure supplement 4*). We hypothesized that such probes either (i) cross-hybridize with the MT genome (i.e. lie within NUMT regions) or (ii) corresponded to genetic loci involved in regulation of mtDNA-CN. As an illustration, *Supplementary file 1* – Tab 1 conveys characteristics of the 10 most strongly correlated variants. Four of the top 10 probes were located within 1 Mb of a NUMT region, and in all 4 cases, the sign of the correlation coefficient was negative which might reflect interference of autosomal signal with increasing mtDNA-CN. An additional three probes corresponded to variants within genes that were implicated in MT disorders or regulation of MT processes (*Supplementary file 1* – Tab 1).

We further explored whether there was evidence for cross-hybridization between autosomal SNPs and sex chromosomes by regressing adjusted autosomal probe intensities with reported male status. Generally, correlations between autosomal intensities and sex were stronger than those with MT intensities, suggesting that cross-hybridization of autosomal probes to sex chromosomes is more pronounced. For the top 10 sex-associated probes, we performed BLASTn alignment against the human reference genome (GRCh38) using 30 bases surrounding each probe (*Supplementary file 1* – Tab 2) (*Altschul et al., 1990*). All probes had at least one flanking sequence with near-perfect ( >97%) sequence identity to a sex chromosome. For these 10 probes, the sign of the correlation between autosomal intensity and male status was perfectly consistent with homology to X or Y chromosomes, thus supporting the hypothesis that such probes cross-hybridized to sex chromosomes.

Inclusion of autosomal probes with evidence of off-target hybridization to the MT genome or sex chromosomes is problematic. In the former scenario, misattribution of autosomal as MT signal may reduce the effectiveness of normalization. In the latter scenario, inadvertent adjustment for sex through retention of cross-hybridizing autosomal probes may occur if such probes explain substantial variance in autosomal probe intensities and this is particularly problematic given that mtDNA-CN has been robustly shown to differ between genders in epidemiological studies. In preliminary investigations where sex-associated probes were retained, we noticed that several top PCs that perfectly tagged gender. Hypothetically, had these PCs been retained and used for correction of MT signal, then the final mtDNA-CN estimate would have been inadvertently corrected for gender. Therefore,

we removed autosomal probes exhibiting at least modest correlation (|r| > 0.05) with reported gender (907; 1.14%) or median MT intensity (193; 0.24%) and then recalculated top autosomal PCs.

## PCA-based approach improves concordance with complementary estimates

After correcting MT probes using the updated set of 120 autosomal L2R PCs, we adopted the Mito-Pipeline's approach for estimating mtDNA-CN and calculated the median of corrected MT L2R values to denote an individual's final mtDNA-CN estimate. Using this median-based approach, array mtDNA-CN estimates demonstrated significant correlation with WES mtDNA-CN estimates ($r = 0.33$; $p<2.23 \times 10^{-308}$). However, we found that performing PCA across all corrected MT L2R values and then extracting the top MT PC for each sample as the final mtDNA-CN estimate resulted in stronger correlation with WES ($r = 0.49$; $p<2.23 \times 10^{-308}$).

## Independent validation of AutoMitoC in the INTERSTROKE study

We additionally validated the AutoMitoC pipeline by deriving array-based mtDNA-CN estimates in the INTERSTROKE study and comparing these with parallel qPCR-based measurements, the current gold standard for measuring mtDNA-CN. INTERSTROKE is an international case-control study of stroke including 26,526 participants from 32 countries and 142 centers (*O'Donnell et al., 2010*). Blood samples have been collected for a subset of approximately 12,000 individuals, of which 9311 have been successfully genotyped using the Axiom Precision Medicine Research Array (PMRA r3). MtDNA-CN estimates for INTERSTROKE were generated according to the AutoMitoC pipeline as illustrated in *Figure 1*. A further subset of 5791 samples with both suitable array genotypes have undergone qPCR measurement of mtDNA-CN using the plasmid-normalized protocol from *Fazzini et al., 2019*. Please see *Chong et al., 2021* for further details *Chong et al., 2021*.

# GWAS in the UKBiobank

## Data acquisition and quality control

UKBiobank samples were genotyped on either the UK Biobank Array (~450,000) or the UK BiLEVE array (~50,000) for approximately 800,000 variants (*Bycroft et al., 2018*). Further imputation was conducted by the UKBiobank study team using a combined reference panel of the UK10K and Haplotype Reference Consortium datasets. Imputed genotypes (version 3) for 488,264 UKBiobank participants were downloaded through the European Genome Archive (Category 100319). Samples were removed if they were flagged for any of the UKBiobank-provided quality control annotations (Resource 531; 'ukb_sqc_v2.txt') for high ancestry-specific heterozygosity, high missingness, mismatching genetic ancestry, or sex chromosome aneuploidy ('het.missing.outliers,' 'in.white.British.ancestry.subset,' 'putative.sex.chromosome.aneuploidy'). Samples were also removed if their submitted gender did not match their genetic sex or if they had withdrawn consent at the time of analysis. Variant quality control consisted of removing variants that had low imputation quality (INFO score $\leq 0.30$), were rare (MAF $\leq 0.005$), or were in violation of Hardy Weinberg Equilibrium (HWE $p \leq 1 \times 10^{-10}$). The HWE test was conducted within a subset of unrelated individuals for each ethnic strata, though all related individuals were retained for subsequent GWAS analysis. Lastly, in special consideration of mtDNA-CN as the GWAS phenotype, we also removed variants within 'NUMTs,' which refer to regions of the nuclear genome that exhibit homology to the MT genome due to past transposition of MT sequences. Accordingly, NUMTs represent a specific confounder of mtDNA-CN GWAS analyses which may lead to false positive associations. NUMT boundaries were obtained from the UCSC NumtS Sequence (numtSeq) track, which is based on the Reference Human NumtS curated by *Simone et al., 2011*. All sample and variant quality control of imputed genotypes were executed using qctools and resultant bgen files were indexed with bgenix. Smaller ethnic groups with similar genetic ancestry were consolidated; individuals self-reporting as 'African' or 'Caribbean' were combined into a larger 'African' stratum and individuals self-reporting as 'Indian' or 'Pakistani' were combined into a 'South Asian' stratum.

## Association testing

GWAS were initially conducted in an ethnicity-stratified manner for common variants (MAF >0.005). The number of variants tested for association with mtDNA-CN varied for British (M = 10,728,525), Irish (M = 10,707,537), other White (M = 10,894,497), South Asian (M = 11,350,981), and African (M = 18,981,896) study participants, respectively. To allow for genetic relatedness between participants,

GWAS were conducted using the REGENIE framework, which consists of two steps (*Mbatchou, 2020*). In step 1, mtDNA-CN was predicted using a ridge regression model fit on a set of high-quality genotyped SNPs (MAF >0.01, minor allele count (MAC) >100, genotype and sample missingness above 10%, and passing HWE [$p > 10^{-15}$]) across the whole genome in blocks of 1000 SNPs. In step 2, the linear regression model was used to test the association of all SNPs adjusting for age, $age^2$, sex, chip type, 20 genetic PCs, and blood cell counts (white blood cell, platelet, and neutrophil counts), and conditional on the model from step 1.

Blood cell counts were determined for blood specimen collected at the initial assessment visit using Beckman Coulter LH750 analyzers (https://biobank.ndph.ox.ac.uk/showcase/showcase/docs/haematology.pdf). Information on blood cell counts was retrieved from the UKBiobank data showcase. Individuals with missing values for any blood cell counts (~2.5%) were removed from any subsequent analysis involving blood cell counts. Quality control of blood counts was done following the same procedure as *Longchamps, 2019* (*Longchamps, 2019*). Except for platelet counts, all blood cell counts were log transformed and samples exhibiting outlying values were removed (~4% samples). Lastly, values were standardized to have a mean of 0 and SD of 1.

After ethnicity-specific GWAS were performed, results were combined through meta-analysis using METAL (*Willer et al., 2010*). European (N = 383,476) and transethnic (N = 395,718) GWAS meta-analyses were performed. Due to the high proportion of Europeans (97%), results from the transethnic meta-analyses strongly resembled that of the European meta-analysis. Accordingly, we report results from the European meta-analysis as the primary GWAS but also provide a comparison of regression coefficients to non-European ethnic groups. To summarize statistical associations, Manhattan plots and quantile-quantile plots were generated by uploading summary statistics into the locus zoom web platform (https://my.locuszoom.org/) (*Pruim et al., 2010*). LD score regression was performed to calculate the LD score intercept by uploading GWAS results to the LDhub test center (http://ldsc.broadinstitute.org/) (*Bulik-Sullivan et al., 2015*). As per the instructions, all variants within the major histocompatibility (MHC) region on chromosome 6 were removed prior to uploading. Annovar (version date June 07, 2020) was used to functionally annotate genome-wide significant loci based on their proximity (±250 kb) to genes (RefSeq), predicted effect on amino acid sequence, allele frequency in external datasets (1000 Genomes), clinical pathogenicity (Clinvar), and in silico deleteriousness (CADD), and eQTL information (GTEx v8) (*1000 Genomes Project Consortium et al., 2012*; *GTEX, 2014*; *Landrum et al., 2016*; *Rentzsch et al., 2019*).

## NUMT sensitivity analyses

To assess whether genome-wide significant associations could be explained by the interference of cryptic MT pseudogenes (NUMTs), we performed sensitivity analyses as per *Nandakumar et al., 2021*. The MT genome was divided into thirds, and AutoMitoC estimates were rederived for each region (MT:1–6425; MT:6526–11947; MT:11948–16569) using the corresponding MT variants belonging to these three consecutive regions. Association testing was repeated for each region using REGENIE. For a given variant, if at least one region-based analysis yielded a nonsignificant association ($p < 0.05$), we considered this as evidence of NUMT interference.

## Fine-mapping of GWAS signals

We followed a similar protocol to *Vuckovic et al., 2020* for fine-mapping mtDNA-CN loci (*Vuckovic et al., 2020*). Genome-wide significant variants were consolidated into genomic blocks by grouping variants within 250 kb of each other. LDstore was used to compute a pairwise LD correlation matrix for all variants within each block and across all samples included in the European GWAS meta-analysis (*Benner et al., 2017*). For each genomic block, FINEMAP was used to perform stepwise conditional regression (*Benner et al., 2016*). The number of conditionally independent genetic signals per genomic block was used to inform the subsequent fine-mapping search parameters. Finally, the FINEMAP random stochastic search algorithm was applied to derive 95% credible sets constituting candidate causal variants that jointly contributed to 95% (or higher) of the posterior inclusion probabilities (*Benner et al., 2016*).

## Mt-eQTL and heteroplasmy look-ups

Among GWAS hits, we searched for mt-eQTLs using information from *Ali et al., 2019*, 'Nuclear genetic regulation of the human mitochondrial transcriptome' (*Ali et al., 2019*). When mt-eQTLs also had reported effect estimates, the consistency in direction-of-effects between mt-eQTL and mtDNA-CN associations was reported. Genome-wide significant variants from a heteroplasmy GWAS by *Nandakumar et al., 2021* were interrogated in the same manner (*Nandakumar et al., 2021*).

## Gene prioritization and pathway analyses

The Data-driven Expression-Prioritized Integration for Complex Traits (DEPICT) v.1.1 tool was used to map mtDNA-CN loci to genes based on shared coregulation of gene expression using default settings (*Pers et al., 2015*). Genome-wide significant variants from the European GWAS meta-analysis were 'clumped' into independent loci using PLINK '--clump-p1 5e-8 --clump-kb 500 --clump-r2 0.05' with LD correlation matrix derived from 1000 Genomes Europeans (*Purcell et al., 2007*). DEPICT was subsequently run on independent SNPs using default settings. DEPICT identified 91 genes in total at an FDR of 0.05. Of the 91 genes, 4 noncoding genes were excluded from subsequent analyses for lack of a match in the GeneMANIA database (*Warde-Farley et al., 2010*). The excluded genes include a pseudogene (*PTMAP3*), an intronic transcript (*ALMS1-IT1*), and two long noncoding RNAs (*SNHG15*, *RP11-125K10.4*). The remaining 87 DEPICT-prioritized genes were uploaded to the Gene-MANIA web platform (https://genemania.org/), which mines publicly available biological datasets to identify additional related genes based on functional associations (genetic interactions, pathways, coexpression, colocalization, and protein domain homology). Based on the combined list of DEPICT and GeneMANIA identified genes, a network was formed in GeneMANIA maximizing the connectivity between all input genes, using the default setting, 'Assigned based on query gene.' Functional enrichment analysis was then performed to identify overrepresented GO terms among all network genes (*Ashburner et al., 2000*). All network genes with at least one GO annotation were compared to a background comprising all GeneMANIA genes with GO annotations.

## MT annotation-based analyses

To complement the previous pathway analyses, we labeled prioritized genes with MitoCarta3 annotations and performed subsequent statistical enrichment analyses (*Rath et al., 2021*). MitoCarta3 is an exquisite database of MT protein annotations, which draws from mass spectrophotometry and green fluorescent protein colocalization experiments of isolated mitochondria from 14 different tissues to assign all human genes statuses indicating whether the corresponding proteins are expressed in the mitochondria or not. We tested whether prioritized genes were enriched for the MT proteome by using a binomial test in R. The number of 'trials' was set to the total number of DEPICT and GeneMANIA-prioritized genes (107); the number of 'successes' was set to the aforementioned gene subset that was labeled as MT proteins by MitoCarta3 (27); finally, the expected probability was set to the number of nuclear-encoded MitoCarta3 genes divided by the total number of genes (1120/18,922). Furthermore, a t-test was used to compare mean PGC-1A induced fold change for the subset of GWAS-prioritized genes expressed in the MT proteome (27) as compared to the mean PGC-1A induced fold change for all 1120 nuclear MitoCarta3-annotated genes. Also, genes were categorized based on MitoCarta3 'MitoPathway' annotations. MitoCarta3 genes with missing values were excluded from this analysis. Lastly, the 27 genes were labeled based on MitoCarta3 'MitoPathways.' Only the top-level pathway (i.e. parent node) was ascribed to each gene.

## Genetic analysis of rare variants

### Data Acquisition and Quality Control

Population-level WES variant genotypes (UKB data field: 23,155) for 200,643 UKBiobank participants corresponding to 17,975,236 variants were downloaded using the gfetch utility. These data represent the second tranche of WES data released by the UKBiobank and differ from the first tranche (~50 K samples) which was used for the development of the AutoMitoC pipeline. Quality control of WES data was conducted as follows. First, 11 samples which withdrew consent by the time of analysis were removed. Second, 83,700 monomorphic variants were removed. Third, 369,215 variants with nonmissing genotypes present in less than 90% of samples were removed. Fourth, two samples with call rates less than 99% were removed. Fifth, 18 samples exhibiting discordance between genetic and

reported sex were removed. Sixth, through visual inspection of scatterplots of the first two genetic PCs, three outlying samples whose locations strongly departed from their putative ethnicity cluster were removed. Seventh, 35,317 variants deviating from HWE were removed. Eighth, 12,765 samples belonging to smaller ethnic groups with less than 5000 samples (South Asian = 3395; African = 3168; other = 6202) were removed. Ninth, we selected for a maximal number of unrelated samples and excluded 14,156 samples exhibiting third degree or closer relatedness. Finally, 12,394,404 noncoding variants were removed, and 5,176,300 protein-altering variants (stopgain, stoploss, startloss, splicing, missense, frameshift, and in-frame indels) were retained in 173,688 samples.

## ExWAS to identify rare mtDNA-CN loci

Of the 173,688 individuals, suitable AutoMitoC mtDNA-CN estimates were available for 147,740 samples. Rare variant inclusion criteria consisted of variants which were infrequent (MAF ≤0.001), nonsynonymous, and predicted to be clinically deleterious by Mendelian clinically applicable pathogenicity v.1.4 scores (or were highly disruptive variant types including frameshift indel, stopgain, stoploss, or splicing) (*Jagadeesh et al., 2016*). Herein, such variants are referred to as 'rare variants' for simplicity. For each gene, rare allele counts were added per sample. A minor allele count threshold of at least 10 rare alleles was applied leading to a total of 18,890 genes analyzed (exome-wide significance $p<0.05/18,890 = 2.65 \times 10^{-6}$). Linear regression was conducted using mtDNA-CN as the dependent variable and the rare alleles counts per gene as the independent variable. The same set of covariates used in the primary GWAS analysis was used in the ExWAS analysis.

## Phenome-wide association testing for rare *SAMHD1* mutation carrier status

To identify disease phenotypes associated with carrying a rare *SAMHD1* mutation, we maximized sample size for phenome-wide association testing by analyzing the larger set of 173,688 WES samples (with or without suitable mtDNA-CN estimates). Disease outcomes were defined using the previously published 'PheCode' classification scheme to aggregate ICD-10 codes from hospital episodes (field ID 41270), death registry (field ID 40001 and 40002), and cancer registry (field ID 40006) records (*Denny et al., 2013*; *Wu et al., 2019*). Logistic regression was applied to test the association of *SAMHD1* mutation carrier status versus 771 PheCodes (phenome-wide significance $p<0.05/771 = 6.49 \times 10^{-5}$) with a minimal case sample size of 300 (*Wei et al., 2017*). The same set of covariates used in the primary GWAS were also employed in this analysis.

## Mendelian randomization analysis

### Disease outcomes

To assess evidence for a causal role of mtDNA-CN on MT disorder-related traits, we first defined a list of testable disease outcomes related to MT disorders. We cross-referenced a list of 36 clinical manifestations of MT disease to FinnGen consortium GWAS (release 4; November 30, 2020) traits (*Gorman et al., 2016*; *Feng, 2020*). Among 2444 FinnGen traits, 10 overlapped with MT disease and had a case prevalence greater than 1% and were chosen for two-sample Mendelian randomization analyses. These 10 traits included type 2 diabetes (N = 23,364), mood disorder (N = 20,288), sensorineural hearing loss (N = 12,550), cerebrovascular disease (N = 10,367), migraine (N = 6687), dementia (5675), epilepsy (N = 4558), paralytic ileus and intestinal obstruction (N = 2999), and cardiomyopathy (N = 2342). FinnGen effect estimates and standard errors were used in subsequent Mendelian randomization analyses to define the effect of selected genetic instruments on disease risk.

### Genetic instrument selection

First, genome-wide significant variants from the present European GWAS meta-analysis of mtDNA-CN were chosen (N = 383,476). Second, we matched these variants to the FinnGen v4 GWAS datasets (*Feng, 2020*). Third, to enrich for variants that directly act through MT processes, we only retained those within 100 kb of genes encoding for proteins that are expressed in mitochondria based on Mito-Carta3 annotations (*Rath et al., 2021*). Fourth, we performed LD pruning in PLINK with 1000 Genomes Europeans as the reference panel to ascertain an independent set of genetic variants (LD r2 <0.01) (*Purcell et al., 2007*; *1000 Genomes Project Consortium et al., 2012*). Lastly, to mitigate potential for horizontal pleiotropy, we further removed variants with strong evidence of acting through alternative pathways by performing a phenome-wide search across published GWAS with Phenoscanner V2

(*Kamat et al., 2019*). Variants strongly associated with other phenotypes ($p < 5 \times 10^{-20}$) were removed unless the variant was a coding mutation located within gene encoding for the MT proteome (Mito-Carta3) or had an established MT role based on manual literature review (*Rath et al., 2021*). Seven genetic variants were removed based on these criteria including rs8067252 (*ADAP2*), rs56069439 (*ANKLE1*), rs2844509 (*ATP6V1G2-DDX39B*), rs73004962 (*PBX4*), rs7412 (*APOE*), rs385893 (*AK3, RCL1*), and rs1613662 (*GP6*). A total of 27 genetic variants were used to approximate genetically determined mtDNA-CN levels. As sensitivity analyses, a broader set of genetic instruments including variants irrespective of their proximity to MT protein-encoding genes was additionally tested.

## Mendelian randomization and sensitivity analyses

Two-sample Mendelian randomization analyses were performed using the 'TwoSampleMR' and 'MR-PRESSO' R packages (*Hemani et al., 2018*; *Verbanck et al., 2018*). Effect estimates and standard errors corresponding to the 27 genetic variants on mtDNA-CN (exposure) and MT disease phenotypes (outcome) were derived from the European GWAS meta-analysis and FinnGen v4 GWAS summary statistics, respectively. Three MR methodologies were employed including inverse variance weighted (primary method), weighted median, and MR-EGGER methods. MR-PRESSO was used to detect global heterogeneity and p-values were derived based on 1000 simulations. If significant global heterogeneity was detected ($p < 0.05$), a local outlier test was conducted to detect outlying SNPs. After removal of outlying SNPs, MR analyses were repeated. In the absence of heterogeneity (Egger-intercept $p \geq 0.05$; MR-PRESSO global heterogeneity $p \geq 0.05$), we reported the inverse-variance weighted result. In the presence of balanced pleiotropy (MR-PRESSO global heterogeneity $p < 0.05$) and absence of directional pleiotropy (Egger-intercept $p \geq 0.05$), we reported the weighted median result. In the presence of directional pleiotropy (Egger-intercept $p < 0.05$), we reported the MR-EGGER result. Also, the Steiger directionality test was employed to ensure that a greater proportion of variance in mtDNA-CN was explained than risk of the outcome. To further examine the robustness of these findings, MR analyses were also conducted using the generalized summary data-based MR (GSMR) method which incorporates HEIDI outlier removal and accounts for the uncertainty in SNP-outcome effect estimates. Finally, to replicate the two-sample MR finding using an independent outcome dataset without UKBiobank participants, we repeated two-sample MR analyses using the International Genomics of Alzheimer's Disease Consortium (2013) GWAS meta-analysis including 17,008 cases and 37,154 controls (*Lambert et al., 2013*).

## Acknowledgements

We want to acknowledge the participants and investigators of the UKBiobank and FinnGen studies.

# Additional information

## Funding

No external funding was received for this work.

## Author contributions

Michael Chong, Conceptualization, Data curation, Formal analysis, Methodology, Validation, Visualization, Writing – original draft, Writing – review and editing; Pedrum Mohammadi-Shemirani, Data curation, Formal analysis, Visualization, Writing – original draft, Writing – review and editing; Nicolas Perrot, Formal analysis, Visualization, Writing – review and editing; Walter Nelson, Conceptualization, Methodology, Software, Visualization, Writing – review and editing; Robert Morton, Conceptualization, Supervision, Visualization, Writing – review and editing; Sukrit Narula, Irfan Khan, Mohammad Khan, Tafadzwa Machipisa, Formal analysis, Writing – review and editing; Ricky Lali, Data curation, Formal analysis, Writing – review and editing; Conor Judge, Conceptualization, Methodology, Writing – original draft, Writing – review and editing; Nathan Cawte, Data curation, Validation, Writing – original draft, Writing – review and editing; Martin O'Donnell, Conceptualization, Data curation, Investigation, Resources, Writing – review and editing; Marie Pigeyre, Loubna Akhabir,

Supervision, Writing – review and editing; Guillaume Paré, Conceptualization, Data curation, Investigation, Methodology, Project administration, Resources, Software, Supervision, Writing – review and editing

**Author ORCIDs**
Michael Chong ⓘ http://orcid.org/0000-0002-0555-4622
Pedrum Mohammadi-Shemirani ⓘ http://orcid.org/0000-0001-6740-7858
Robert Morton ⓘ http://orcid.org/0000-0003-0099-4167
Guillaume Paré ⓘ http://orcid.org/0000-0002-6795-4760

**Ethics**
**Ethics Statement:**Approval was received to use UKBiobank study data in this work under application ID # 15,255 ("Identification of the shared biological and sociodemographic factors underlying cardiovascular disease and dementia risk"). The UKBiobank study obtained ethics approval from the North West Multi-centre Research Ethics Committee which encompasses the UK (REC reference: 11/NW/0382). All research participants provided informed consent.

**Decision letter and Author response**
Decision letter https://doi.org/10.7554/eLife.70382.sa1
Author response https://doi.org/10.7554/eLife.70382.sa2

---

## Additional files

**Supplementary files**
• Supplementary file 1. Characteristics of autosomal probes exhibiting strong cross-genome correlation. Table 1. Characteristics of top 10 autosomal probes whose background-corrected L2R values remain correlated with median MT L2R. Table 2. Characteristics of top 10 autosomal probes whose background-corrected L2R values remain associated with gender.

• Supplementary file 2. Supplementary results for primary and secondary GWAS-based analyses.

• Transparent reporting form

**Data availability**
Individual-level UKBiobank genotypes and phenotypes can be acquired upon successful application (https://bbams.ndph.ox.ac.uk/ams/). All individual-level UKBiobank data was accessed as part of application # 15255. FinnGen summary statistics are freely available to download (https://www.finngen.fi/en/access_results). All data products generated as part of this study will be made publicly accessible. Specifically, the AutoMitoC array-based mtDNA-CN estimation pipeline is available on GitHub (https://github.com/GMELab/AutoMitoC, copy archived at swh:1:rev:5669ae00011ae-ce6ed494a3fde241ab3027c59e2). The mtDNA-CN estimates derived in UKBiobank participants have been returned to the UKBiobank and made accessible to researchers through the data showcase (https://biobank.ndph.ox.ac.uk/showcase/). Summary-level association statistics from GWAS have been made publicly available for download from the GWAS catalogue (https://www.ebi.ac.uk/gwas/publications/35023831). All remaining data are available in the main text or supplementary materials.

The following dataset was generated:

| Author(s) | Year | Dataset title | Dataset URL | Database and Identifier |
|---|---|---|---|---|
| Chong M, Mohammadi-Shemirani P, Perrot N, Nelson W, Morton R, Narula S, Lali R, Khan I, Khan M, Judge C, Machipisa T, Cawte N, O'Donnell M, Pigeyre M, Akhabir L, Pare G | 2022 | UKbiobank mtDNA-CN GWAS summary statistics | https://www.ebi.ac.uk/gwas/publications/35023831 | GWAS Catalogue, 35023831 |

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
