## [Editor Report]

This is an original human GWAS study that treats mitochondrial copy number variation as a trait, and investigates its genetic basis, as well as its association with (and possible causal role in) various human diseases, such as cancer and dementia. The study identifies 71 significant loci, show that these are significantly over-represented in a priori candidates, and argue convincingly that this could help us understand how mitochondrial copy number is regulated at a cellular level.

---

## [Decision Letter]

**Decision letter after peer review:**

Thank you for submitting your article "GWAS and ExWAS of blood Mitochondrial DNA copy number identifies 71 loci and highlights a potential causal role in dementia" for consideration by *eLife*. Your article has been reviewed by 3 peer reviewers, including Magnus Nordborg as the Reviewing Editor and Reviewer #1, and the evaluation has been overseen by Y M Dennis Lo as the Senior Editor.

Essential revisions:

1. Your method for estimating mtDNA copy number needs to be better described and explored. We need: a more detailed description of the approach; a more thorough benchmarking (comparing to standard approaches and against estimates from sequencing data); a more thorough description of the phenotype and its distribution; comparisons with age, sex and other UK Biobank phenotypes; description of samples removed and the rationale for removing them.

2. More thorough Mendelian Randomisation analysis: test for pleiotropy using Heidi; perform comparative analysis using all genome wide significant hits (with accompanying discussion of any differing results).

3. Place work in context of other two highly similar recent studies (Longchamps 2021 and Hagg 2021).

4. Data release: code should be on GitHub, raw data for mtDNA-CN estimates and covariates provided where possible, summary statistics uploaded to GWAS catalogue.

5. A bit more discussion of potential population structure confounding? You focus is on individuals of European decent, with meta-analysis including non-Europeans giving similar results.

*Reviewer #1:*

This is an original human GWAS study that treats mitochondrial copy number variation as a trait, and investigates its genetic basis, as well as its association with (and possible causal role in) various human diseases. The main strength of the study is the development of a method for estimating mitochondrial number from standard array data. This makes it possible to reuse the vast UK Biobank data (n = 383,476) and work with much larger sample sizes than previous studies. The estimated phenotypes are sensibly QC-ed using various data.

They then carry out a standard GWAS, identify 71 significant loci, show that these are significantly over-represented in a priori candidates, and argue convincingly that this could help us understand how mitochondrial copy number is regulated at a cellular level. They show that rare SAMHD1 variants associated with high copy number are also associated with increased cancer risk, and, finally, that mitochondrial copy number may play a role in dementia.

The study represents clever re-use of data, and the findings appear to be well supported and should be interesting to a broad audience. Unlike many human GWAS, there is clearly real biology here.

Your analyses appear admirably solid to me, but this need to be verified by an expert on human GWAS. The same is true for the novelty of the results.

The only improvement I would like to see lies in the description of the estimated phenotype. There is little detail about its distribution, and I would also like to learn more about the factors influencing it, including, obviously, an estimate of much of the variation your 71 loci explain, and a discussion of what explains the rest of the variation. The UK Biobank is full of data…

*Reviewer #2:*

Mitochondrial DNA copy number is a significant aspect of mitochondrion function, both of which have been linked to disease states. This is an interesting assessment of the genetic basis for variation in mitochondrial DNA copy number. The authors describe a new method for mtDNA CN estimation, although additional benchmarking and validation is warranted in this reviewer's opinion. The GWAS/ExWAS analyses are fairly standard and identify genetic variants likely influencing mtDNA CN. The implementation of the MR analyses is more superficial; it remains unclear in this reviewer's opinion that pleiotropy has been effectively ruled out. The estimates of mtDNA CN and other non-other non-identifiable information (eg, sex, age, ethnic group, etc) necessary to replicate the author's findings should be made publicly available as easily accessible supplementary files.

1. Regarding the method for mtDNA estimation. The method is barely described in the main body of the manuscript. More information should be provided in the main manuscript given that the method it is so essential for this work. Greater clarity about the reproducibility would have helped. Also, it appears that the estimates of mtDNA CN were not tested against a standard with known CN, could this be performed? Some benchmark was done with qPCR estimates, while the results of the methods here developed and qPCR are positively correlated, the magnitude of the correlation is moderate. Finally, wow does the new method compare with sequencing based estimates? All in all, more benchmarking of the method in the main manuscript would greatly improve the paper.

2. Besides better description of the methods for mtDNA CN estimation, a much more detailed description of the mtDNA CN data should be provided in the main manuscript. It is not possible to properly understand/appreciate the reported associations and MR results without such description of the underlying mtDNA CN data.

3. In this reviewer's opinion the pre-computed mtDNA estimates and other non-identifiable information (eg, sex, age, ethnic group, etc) necessary to replicate the author's findings should be made publicly available in supplementary text format files and that could be easily accessible to readers. If the text files are too large for *eLife* the authors should upload the files to Dryad, dbGaP, etc.

4. The author's state that "After quality control, 359,689 British, 10,598 Irish, 13,189 Other White, 6,172 South Asian, and 6,133 African samples had suitable array-based mtDNA-CN estimates for subsequent GWAS testing. It is unclear what the authors mean by "suitable" in this case. It would also be good to describe in the main text how many were unsuitable in each case and the reasons why these estimates did not match the requirement for suitability.

5. It is unclear if the authors detected ethnic and/or population differences that might confound the other downstream analyses. It is also unclear what are the error and reproducibility of the CN estimates.

6. The influence of sex and age are not described (or superficially described).

7. The presentation of the MR results is unconvincing. For the analysis, the authors have not ruled out pleiotropy in my opinion. Heterogeneity tests such as Heidi should be used Zhu, Z. et al. Integration of summary data from GWAS and eQTL studies predicts complex trait gene targets. Nat. Genet. 48, 481-487 (2016) in addition to MREgger. Also, dementia seems to have the largest error bars around its OR estimates. More detailed analyses and discussion could help here.

*Reviewer #3:*

This work develops and presents an adapted approach to quantify mitochondrial DNA copy number (mtDNA-CN) from DNA genotyping arrays (named AutoMitoC) by using the intensity of probes emanating from the mitochondrial genome and then using a series of control and normalisation steps that can be applied within a standard framework across data from different populations. They use their estimates of mtDNA-CN as a quantitative trait within a genome wide association study to identify genes and genetic variants that are associated with variation in this phenotype, using both rare and common polymorphisms. Through multiple complementary approaches they then identify a large number of genes that are linked to mtDNA-CN, including those that are involved in mtDNA depletion disorders and various different components of mitochondrial function. Within this, using rare variant analysis they find SAMHD1 as a potential regulator of mtDNA-CN (and possibly breast cancer risk), and the authors suggest that the product of this gene may make a good draggable target to control the downstream effects of altered mtDNA-CN. Finally, the study uses Mendelian Randomisation analyses between genetically determined mtDNA-CN and a selection of disease phenotypes that have been linked to mitochondrial processes previously, and in doing so they find a link between mtDNA-CN and dementia.

In general, the work appears robust and achieves the aims of uncovering the genetic architecture of a disease relevant phenotype, and its potential downstream implications. Readers of this article should be aware of two further studies that have performed similar analyses on largely the same data as those described in this paper. The first by Hagg et al. (2021) considers largely the same set of individuals and identifies 50 genetic loci associated with variation in mtDNA-CN across individuals. The second by Longchamps et al. (2021), finds 129 independent genetic variant associations and includes additional data from a different cohort. The study presented here discusses the findings of Hagg et al. (although a direct comparison of genes is not given) but does not interrogate and compare results from the Longchamps study, which is necessary to truly understand the genetic architecture of this trait and to place this work in context. For the comparison with Hagg et al., the authors suggest that their improved method for mtDNA-CN quantification leads to a higher number of genetic associations, although a more formal comparison of AutoMitoC and standard approaches against control data like that generated from qPCR would make this statement more robust (there is a comparison between AutoMitoC and Hagg et al. quantification in the Discussion section, although it is not clear whether this comparison is like-for-like using the same set of samples). In general, many of the bigger picture biological findings are shared across the three studies (many shared loci and enrichment of genes involved in mtDNA depletion syndrome and mitochondrial processes), which is an excellent advert for reproducible science.

The additional strength of the work by Kong et al. are two extra pieces of analyses not found in other works – an analysis of rare variants and the implementation of Mendelian Randomisation to link variation in mtDNA-CN to disease risk. For the Mendelian Randomisation analyses, authors select genetic variants associated with mtDNA-CN based on their likely role in mitochondrial processes (those variants falling within or close to a mitocarta gene). This is rational given the assumption in Mendelian Randomisation analyses, that genetic instruments should be causal, but it would also be useful to also consider the full set of genetic variants that are associated with mtDNA-CN at genome wide significance to test the relevance of selecting a subset of genetic variants.

References:

RJ Longchamps et al., Genetic analysis of mitochondrial DNA copy number and associated traits identifies loci implicated in nucleotide metabolism, platelet activation, and megakaryocyte proliferation, and reveals a causal association of mitochondrial function with mortality. bioRxiv 2021.01.25.428086; doi: https://doi.org/10.1101/2021.01.25.428086

Hägg S, Jylhävä J, Wang Y, Czene K, Grassmann F. Deciphering the genetic and epidemiological landscape of mitochondrial DNA abundance. Hum Genet. 2021 Jun;140(6):849-861. doi: 10.1007/s00439-020-02249-w.

1. Page 3, lines 3-8 (but also throughout the text): Whilst it is appreciated that there is a vast body of literature pointing to links between mtDNA-CN and complex disease, I think it is worth briefly discussing caveats to these results in order to properly frame the relevance of the work. For instance, there are many studies showing that links between blood derived mtDNA-CN and age-related diseases may at least in part be driven by cell type composition. There is also plenty of debate around whether mtDNA-CN in blood is indicative of processes in other tissues which might be more relevant for each particular disease. Much of these caveats have been recently neatly summarised in Picard 2021.

2. Page 9, lines 16-17: "We postulated that mtDNA-CN loci may regulate copy number by inducing changes in expression of genes that are directly transcribed from mtDNA.": I don't understand this rationale – are you postulating that lower/higher expression from the MT genome may trigger changes in copy number (and thus these may be modulated by variants influencing expression)? If so, this should be stated more clearly, if not, please explain the thinking here. I also think that it is worth rewording your conclusions from this analysis, as there is no test of directionality.

3. Page 9: lines 31-41: Similar to point 2, it would be good to be a clear rationale for the comparison of loci associated with mtDNA-CN and heteroplasmy level.

4. Page 10: lines 29-35: It would be good to see statistical tests comparing the fractions of genes that fall into the mitochondrial process categories described here, versus proportions of genes in these categories in Mitocarta as a whole – are these categories enriched for genes identified in this study?

5. Page 25: lines 20-31: Links should be provided in any revised manuscript, including the GitHub repository for the method, and summary statistics should be uploaded to the GWAS catalogue.

6. Figures: In general, the figures are not particularly clear and should be improved for publication. For example, axes text is often too small and difficult to read, and text labels on figures 1 and 2 are not aligned properly.

References:

Picard, M., Blood Mitochondrial DNA Copy Number: What Are We Counting? Mitochondrion (2021), doi: https://doi.org/10.1016/j.mito.2021.06.010

---

## [Author Response]

Essential revisions:1. Your method for estimating mtDNA copy number needs to be better described and explored. We need: a more detailed description of the approach; a more thorough benchmarking (comparing to standard approaches and against estimates from sequencing data); a more thorough description of the phenotype and its distribution; comparisons with age, sex and other UK Biobank phenotypes; description of samples removed and the rationale for removing them.

We wholly agree that a more detailed description of the method in the main text is warranted to improve the readability of the manuscript. We apologize for not realizing that the Materials and methods section does not contribute to the manuscript word count, hence why we originally provided only a general overview of the method in the main manuscript. Accordingly, we have added more detail to nearly all sections in the Materials and methods.

To specifically address the issue of further clarity regarding the AutoMitoC method, we added a figure describing the methodology step-by-step (Results, page 5, lines 1-21; page 6, lines 1-4).

We also provide an extensive description of the rationale behind AutoMitoC and details on how it was developed in the UKBiobank by adding a new subsection to the Materials and methods subheading “The Automatic Mitochondrial Copy (AutoMitoC) Number Pipeline”.

With respect to more clarity on sample exclusions, further details have now been added to the Materials and Methods section, for the GWAS:

“Initial quality control of 488,264 samples and 784,256 directly genotyped variants was executed in PLINK followed that recommended by the MitoPipeline (i.e. sample call rate > 0.96; variant call rate > 0.98; HWE p-value > 1x10^-5^; PLINK mishap P-value > 1x10^-4^; genotype association with sex p-value > 0.00001; LD-pruning r2 < 0.30; MAF > 0.01) (Purcell et al., 2007). Variants within 1 Mb of immunoglobulin, T-cell receptor genes, and centromeric regions were removed. After this quality control procedure, 466,093 samples and 86,677 common variants remained. Next, genomic waves were corrected according to Diskin et al. (2008) using the PennCNV “genomic_wave.pl” script (https://github.com/WGLab/PennCNV/blob/master/genomic_wave.pl) (Wang et al., 2007; Diskin et al., 2008). Samples with high genomic waviness (L2R SD > 0.35) before and after GC-correction were removed resulting in 431,501 samples with array L2R values corresponding to 86,677 common autosomal variants. Lastly, we excluded samples representing blood cell count outliers (> 3SDs) as per Longchamps et al. (2019) which led to 395,781 participants (Longchamps, 2019). Finally, we took the intersection of European samples with both suitable array and whole-exome sequencing data resulting in a final testing dataset of 34,436 European participants. To evaluate the possibility of replacing common autosomal variant signal normalization with rare variants, we also analyzed a set of 79,611 rare variants with a MAF<0.01. ”

Exome sequencing analysis:

“Population-level whole-exome sequencing (WES) variant genotypes (UKB data field: 23155) for 200,643 UKBiobank participants corresponding to 17,975,236 variants were downloaded using the gfetch utility. […] Finally, 12,394,404 non-coding variants were removed, and 5,176,300 protein-altering variants (stopgain, stoploss, startloss, splicing, missense, frameshift and in-frame indels) were retained in 173,688 samples.”

Regarding the request for further details about the mtDNA-CN phenotype and its determinants by the Reviewers, we performed an analysis of key determinants of mtDNA-CN including age, sex, ethnicity, and blood cell traits and presented these in the Results section:

“In the larger UKBiobank dataset of 395,781 with suitable array-based estimates, the distribution of mtDNA-CN was approximately normal (Figure 2A). […] Collectively, total white blood cell, platelet, and neutrophil counts explained 12.3% variance in mtDNA-CN levels.”

In response to Reviewer #2’s recommendation to perform further benchmarking against sequencing datasets, we attempted to compare array-based estimates to WES-based estimates in a subset of 412 INTERSTROKE participants who had been exome sequenced. Unfortunately, the exome capture method strongly depleted for the mitochondrial genome thus leading to insufficient coverage to make mtDNA-CN estimates. Furthermore, we hope the reviewers will appreciate that not only will additional experimentation require more time and costs, but also INTERSTROKE samples have undergone extensive molecular testing and thus limited DNA quantities remain for these precious samples. Moreover, we note that our method has been benchmarked to the same, if not higher, standard than other array-based methods. Whereas AutoMitoC was compared to both qPCR and exome sequencing-based estimates in two independent datasets, the MitoPipeline was benchmarked against qPCR only, and the method by Hagg et al. (2021) was benchmarked against whole-exome sequencing-based calls in the UKBiobank only. To our knowledge, neither competing method has been benchmarked in different ethnic groups individually as we had done, so the extensibility of their methods outside of European participants remains unproven. Granted, we agree that further benchmarking could be performed and thus have amended the manuscript Discussion (page 18, lines 25-29): accordingly:

“AutoMitoC was benchmarked using multiple comparative mtDNA-CN estimation modalities, in independent cohorts, and in various ethnic groups, but further work can be done to assess the reproducibility and error of mtDNA-CN estimates, calibration to absolute counts, and the robustness to different genotyping platforms.”

2. More thorough Mendelian Randomisation analysis: test for pleiotropy using Heidi; perform comparative analysis using all genome wide significant hits (with accompanying discussion of any differing results).

We greatly appreciate the suggestion to perform additional sensitivity MR analyses. Accordingly, we have repeated MR analyses using the GSMR method which not only incorporates HEIDI outlier removal (HEIDI P < 0.05) but also accounts for the uncertainty in SNP-outcome effect estimates (unlike IVW, weighted median, and MR-EGGER) to improve statistical power. GSMR yielded highly consistent results to our previous analyses; among the 10 phenotypes tested, only dementia was statistically significant (OR=1.95; 95% CI, 1.34-2.84; P=0.0005). This effect estimate was highly concordant with that previously obtained from IVW (OR=1.93; 95% CI, 1.32-2.84; p=0.00008). Furthermore, as per the reviewer’s recommendation, we also performed GSMR using the broader set of genome-wide significant SNPs (irrespective of being located proximal to or within a mitochondrial protein-encoding gene). The association with dementia remained directionally concordant and statistically significant (OR=1.31; 95% CI, 1.02-1.68; P=0.035). Consequently, we have amended the Results section (page 15, lines 17-23) to reflect this addition:

“Findings were robust across several different MR methods including the Weighted Median (OR=2.47; 95% CI, 1.93-3.00; P=0.001), MR-EGGER (OR=2.41; 95% CI, 1.71-3.11; P=0.02), and GSMR-HEIDI (OR=1.95; 95% CI, 1.34-2.84; P=0.001) methods. Results also remained statistically significant when using a broader set of genetic instruments including all genome-wide significant loci irrespective of whether variants were located proximally to MitoCarta3 genes (GSMR-HEIDI OR=1.31; 95% CI, 1.02-1.68; P=0.04).”

Regarding the reviewer’s comment about larger confidence intervals for dementia, this is likely explained by lower prevalence of dementia cases as compared to other phenotypes assessed. Although the replication of this result using an independent set of outcome summary statistics provides further support that this association is unlikely due to chance alone, we further acknowledge this important limitation in the Discussion (page 18, lines 40-42; page 19, lines 1-3).

“Fourth, Mendelian Randomization analyses were underpowered to conduct a broad survey of diseases in which mitochondrial dysfunction may play a causal role, and even among the ten phenotypes tested, statistical power varied by disease prevalence as evidenced by relatively large confidence intervals for the dementia association. Accordingly, future studies including larger case sample sizes are necessary to provide more precise effect estimates.”

3. Place work in context of other two highly similar recent studies (Longchamps 2021 and Hagg 2021).

We thank the reviewers and editorial team for raising this very important consideration. The major ways in which our work distinguishes itself from the other studies is as follows:

1. AutoMitoC represents a methodological advance in array-based mtDNA-CN estimation over the MitoPipeline and the novel method proposed by Hagg *et al.* (2021) in several aspects summarized below:

a. Superior performance as quantified by correlation coefficient with alternative mtDNA-CN measurement techniques:

i. qPCR-based (MitoPipeline R=0.51 vs. AutoMitoC R=0.64)

ii. WES-based (Hagg R=0.33 vs. AutoMitoC R=0.45)

b. Ease-of-implementation, performance, and evaluation in ethnically diverse studies

c. The removal of problematic probes is streamlined through empirical filtering as opposed to manual curation which is time-consuming and subjective (as in the MitoPipeline).

d. Unlike the method by Hagg *et al.* but in the same manner as the MitoPipeline, AutoMitoC only necessitates array genotyping data to derive mtDNA-CN estimates. In contrast, Hagg *et al.* also requires secondary molecular testing (e.g. exome sequencing as in Hagg *et al.*) in a subset of array-genotyped samples for calibration purposes.

e. AutoMitoC has undergone more intensive benchmarking than either method including comparison to mtDNA-CN estimates from two different modalities and in two independent datasets (as mentioned above in the response to Essential Revision #2).

2. We undertook additional sensitivity testing to verify the robustness of identified genetic associations to NUMT interference. Following the scheme developed by Nandeyakumar *et al.* (2021), we rederived mtDNA-CN estimates using sliding window tertiles of MT SNPs and then repeated the GWAS to assess whether genetic associations remained consistent across the three mitochondrial genome segments. If genetic signals were specific to a single region, then this may suggest that NUMTs may underlie the association. Notably, such sensitivity analyses were not performed by either group which leaves opens the possibility that some of these reported associations may be driven by NUMTs.

3. As acknowledged by Reviewer #3, we present several unique downstream analyses not present in the other works:

a. Mendelian Randomization analyses to assess contexts where genetically determined mtDNA-CN levels may influence disease risk

b. Rare variant analyses to identify genes in which rare protein-altering variation impacts mtDNA-CN levels.

c. Phenome-wide association testing for rare variants in *SAMHD1* to understand associated phenotypic consequences.

We also performed a direct comparison of the genome-wide significant associations identified by all three studies by searching hits from the other two studies in our own GWAS analyses and is now reflected in the Results section (page 9, lines 1-31):

“We also compared our findings to genome-wide significant results from two recent GWAS of mtDNA-CN by Hagg et al. (2020) and Longchamps et al. (2021) to better contextualize our findings. […] Notably, these 11 associations were also not reported in Hagg et al. (2020).”

We also note that the work by Hagg *et al.* (2020) has gone through peer review and is published by *Human Genetics*, whereas to our knowledge, the work by Longchamps *et al.* (2021) remains in pre-print. Given that the findings from Longchamps *et al.* (2021) has not gone through due process (albeit appears to be of excellent quality), we hope the reviewers and editorial team can understand our decision to focus the discussion of the comparison of papers on the peer-reviewed work by Hagg *et al.,* out of respect for the peer review process*.* Granted, our work will still add to the mtDNA-CN narrative by Longchamps *et al.* (2021) when published for the aforementioned reasons. Accordingly, we have amended the following Discussion paragraph (page 16, lines 17-39) comparing our work to Hagg *et al.* (2020):

“While several investigations for mtDNA-CN have been published, the present study represents the most comprehensive genetic assessment published to date(Cai et al., 2015; Longchamps, 2019; Hägg et al., 2020). […] Finally, in the present study we included complementary explorations of the role of rare variants through ExWAS and PheWAS, as well as, Mendelian Randomization analyses to assess disease contexts whereby mtDNA-CN may represent a causal mediator and a potential therapeutic target.“

Lastly, given the stronger focus of the revised manuscript on the methodology and as a way to further differentiate our work, with the permission of the editorial team and reviewers, we propose to amend the title of the work to the following: "AutoMitoC: Novel pipeline to estimate mitochondrial DNA copy number from genotyping arrays identifies 71 loci and potential role in dementia."

4. Data release: code should be on GitHub, raw data for mtDNA-CN estimates and covariates provided where possible, summary statistics uploaded to GWAS catalogue.

We have released the code on Github (https://github.com/GMELab/AutoMitoC). An example dataset comprising an anonymized subset of the validation samples including age, sex, and qPCR-based mtDNA-CN estimates is made publicly available for testing.

Raw data for mtDNA-CN estimates for the 395,781 participants have been returned to the UKBiobank data show case (as per the screenshots below) and can be accessed by any registered UKBiobank researcher.

Full genome-wide summary statistics have been uploaded to the GWAS catalogue and are available. The Data and Materials availability statement has been updated to reflect the current statuses:

5. A bit more discussion of potential population structure confounding? You focus is on individuals of European decent, with meta-analysis including non-Europeans giving similar results.

We undertook several precautions to mitigate confounding from population structure (and substructure). Firstly, in all GWAS analyses, association testing was initially performed within each subpopulation separately (British, Irish, Other Caucasian). These stratified analyses limit the potential for SNPs associated with British ancestry, for example, to drive GWAS associations with mtDNA-CN if mtDNA-CN levels systematically differ between subpopulations. Notably, most UKB-based GWAS analyses consolidate these European subpopulations together which would render it more susceptible to this exact type of bias, which is why we opted for this stratified approach. Secondly, to address issues of population substructure within these ethnic groups, we also adjusted for a relatively high number (20) of intra-ethnic genetic principal components. Thirdly, to mitigate the risk of confounding from inter-ethnic differences in downstream analyses as much as possible (e.g. Mendelian Randomization analysis), we also restricted the use of mtDNA-CN GWAS summary statistics to those derived from European participants only to better match the predominantly European ancestry comprising external databases. Lastly, population structure confounding typically manifests as a systematic and strong inflation in genome-wide P-values which was not observed (LD-score intercept=1.036 which is close to the null expectation of 1.00). Accordingly, we added the following to the Discussion (page 18, lines 34-40):

“Third, we did not detect strong evidence of confounding from population structure (or substructure) in our genetic analyses, and we undertook several precautions to mitigate such biases including (i) stratified GWAS analyses by ethnicity, (ii) adjustment for a large number of intra-ethnic principal components, and (iii) restriction of downstream analyses to use of European GWAS summary statistics to harmonize with external databases, however, it is still plausible that population substructure may have influenced our findings.”

Reviewer #1:[…]The only improvement I would like to see lies in the description of the estimated phenotype. There is little detail about its distribution, and I would also like to learn more about the factors influencing it, including, obviously, an estimate of much of the variation your 71 loci explain, and a discussion of what explains the rest of the variation. The UK Biobank is full of data…

We greatly thank the reviewer for these comments and the suggestion. Most of the reviewer’s comments have been relayed as part of Essential Revision #1. Accordingly, please see our detailed response to Essential Revision #1.

As per the reviewer’s additional suggestion to mention the amount of variance explained by identified loci, we have now commented on this in the Results section.

Common loci (page 8, lines 20-21): “These 80 independent genetic signals explained 1.48% variance in mtDNA-CN levels”.

Rare loci (page 13, lines 17-18): “Rare variants in SAMHD1 and TFAM accounted for 0.06% of the variance in mtDNA-CN levels. Collectively, rare and common loci accounted for 1.55%.”

Reviewer #2:Mitochondrial DNA copy number is a significant aspect of mitochondrion function, both of which have been linked to disease states. This is an interesting assessment of the genetic basis for variation in mitochondrial DNA copy number. The authors describe a new method for mtDNA CN estimation, although additional benchmarking and validation is warranted in this reviewer's opinion. The GWAS/ExWAS analyses are fairly standard and identify genetic variants likely influencing mtDNA CN. The implementation of the MR analyses is more superficial; it remains unclear in this reviewer's opinion that pleiotropy has been effectively ruled out. The estimates of mtDNA CN and other non-other non-identifiable information (eg, sex, age, ethnic group, etc) necessary to replicate the author's findings should be made publicly available as easily accessible supplementary files.1. Regarding the method for mtDNA estimation. The method is barely described in the main body of the manuscript. More information should be provided in the main manuscript given that the method it is so essential for this work. Greater clarity about the reproducibility would have helped. Also, it appears that the estimates of mtDNA CN were not tested against a standard with known CN, could this be performed? Some benchmark was done with qPCR estimates, while the results of the methods here developed and qPCR are positively correlated, the magnitude of the correlation is moderate. Finally, wow does the new method compare with sequencing based estimates? All in all, more benchmarking of the method in the main manuscript would greatly improve the paper.

We thank the reviewer for their important comments regarding better description and benchmarking of the AutoMitoC method. This concern was relayed as Essential Revision #1 by the Editorial team. Accordingly, please see our response above to Essential Revision #1.

2. Besides better description of the methods for mtDNA CN estimation, a much more detailed description of the mtDNA CN data should be provided in the main manuscript. It is not possible to properly understand/appreciate the reported associations and MR results without such description of the underlying mtDNA CN data.

The reviewer raises an excellent point which was relayed by the editorial team as a component of Essential Revision #1. Accordingly, please see our response above to this Essential Revision #1.

3. In this reviewer's opinion the pre-computed mtDNA estimates and other non-identifiable information (eg, sex, age, ethnic group, etc) necessary to replicate the author's findings should be made publicly available in supplementary text format files and that could be easily accessible to readers. If the text files are too large for eLife the authors should upload the files to Dryad, dbGaP, etc.

Unfortunately, we are prohibited by the UKBiobank to share individual-level data even if de-identified. The UKBiobank Material Transfer Agreement states that, “The Applicant shall not share, sub-license, disclose, transfer, sell, gift or supply the Materials to any other person or unauthorised third party” (https://www.ukbiobank.ac.uk/media/5cclro0y/applicant-mta-data-only-2021.pdf). However, we have returned the derived individual-level data to the UKBiobank data showcase to enable other approved UKBiobank researchers to access mtDNA-CN estimates. For further details, please see our above response to “Essential Revision #4”. Also, an example dataset derived from our validation cohort with age, sex, and qPCR mtDNA-CN measurements are made available through our GitHub page (https://github.com/GMELab/AutoMitoC ).

4. The author's state that "After quality control, 359,689 British, 10,598 Irish, 13,189 Other White, 6,172 South Asian, and 6,133 African samples had suitable array-based mtDNA-CN estimates for subsequent GWAS testing. It is unclear what the authors mean by "suitable" in this case. It would also be good to describe in the main text how many were unsuitable in each case and the reasons why these estimates did not match the requirement for suitability.

Thank you very much for this suggestion which was relayed by the editorial team as part of Essential Revision #1. As such, we have provided a description of sample exclusion scheme in the Materials and methods section of the main manuscript. For further details, please see above our response to Essential Revision #1.

5. It is unclear if the authors detected ethnic and/or population differences that might confound the other downstream analyses. It is also unclear what are the error and reproducibility of the CN estimates.

We thank the reviewer for pointing out this important confounder. As such, we have further explored the impact of ethnicity on mtDNA-CN levels and added the following to the Results (page 6, lines 35-37):

“MtDNA-CN levels also differed between ethnicities with South Asians (β=-0.18; 95% CI, -0.16 to -0.21; P=2.93x10^-47^) and Africans (β=-0.18; 95% CI, -0.16 to -0.21; P=8.39x10^-48^) having significantly lower levels than Europeans.”

Notably, in the independent INTERSTROKE dataset with qPCR-based mtDNA-CN measurements, we replicated the lower AutoMitoC-based mtDNA-CN levels observed in South Asians (β=-0.18; 95% CI, -0.33 to -0.03; P=0.02) and Africans (β = -0.11; 95% CI, -0.20 to -0.02; P=0.01) relative to Europeans.

To mitigate potential for detecting spurious associations due to inter-ethnic differences, we employed several precautions including (i) performing stratified GWAS analysis within each ethnic group, (ii) adjustment for the top 20 intra-ethnic genetic principal components to capture population substructure within ethnic groups, and (iii) the use of ancestrally matched datasets for downstream analyses. Further details are available above in our response to Essential Revision #5.

To the reviewer’s latter point, indeed, the error and reproducibility of AutoMitoC was not evaluated, and accordingly, we have acknowledged this limitation in the Discussion (page 18, lines 27-29):

“… further work can be done to assess the reproducibility and error of mtDNA-CN estimates, calibration to absolute counts, and the robustness to different genotyping platforms.”

6. The influence of sex and age are not described (or superficially described).

The reviewer raises an excellent point which was relayed by the editorial team as a component of Essential Revision #1. Accordingly, please see our response above to this essential revision.

7. The presentation of the MR results is unconvincing. For the analysis, the authors have not ruled out pleiotropy in my opinion. Heterogeneity tests such as Heidi should be used Zhu, Z. et al. Integration of summary data from GWAS and eQTL studies predicts complex trait gene targets. Nat. Genet. 48, 481-487 (2016) in addition to MREgger. Also, dementia seems to have the largest error bars around its OR estimates. More detailed analyses and discussion could help here.

We thank the reviewer for their suggestions to further scrutinize the robustness of the MR findings with alternative methods and sensitivity analyses. These important considerations were relayed by the editorial team as “Essential Revision #2”. Accordingly, please see our response above to Essential Revision #2.

Reviewer #3:[…]1. Page 3, lines 3-8 (but also throughout the text): Whilst it is appreciated that there is a vast body of literature pointing to links between mtDNA-CN and complex disease, I think it is worth briefly discussing caveats to these results in order to properly frame the relevance of the work. For instance, there are many studies showing that links between blood derived mtDNA-CN and age-related diseases may at least in part be driven by cell type composition. There is also plenty of debate around whether mtDNA-CN in blood is indicative of processes in other tissues which might be more relevant for each particular disease. Much of these caveats have been recently neatly summarised in Picard 2021.

We tremendously appreciate the reviewer’s comment that the positioning of mtDNA-CN as a mitochondrial biomarker was too simplified and should have been more nuanced. Also, thank you very much for providing the very informative reference by Picard (2021). Accordingly, we have better framed the relevance of the work by changing the language around mtDNA as a mitochondrial biomarker throughout the text and by adding the following exposition to the Introduction (page 3, lines 20-37):

“… In contrast to marked drops in mtDNA-CN by 60-80% as seen in those with rare mtDNA depletion syndromes, the relevance of subtler perturbations in mtDNA-CN in disease risk remain to be determined (Basel, 2020).

Granted, the connection between blood mtDNA-CN and aspects of mitochondrial biology remains unclear with many studies showing only moderate correlation between mtDNA-CN and markers of mitochondrial function or biogenesis (Frahm et al., 2005; Wachsmuth et al., 2016). Further complicating the interpretation of epidemiological associations between blood mtDNA-CN and disease risk is the fact that (i) blood mtDNA-CN is strongly confounded by blood cell composition, particularly, platelets that are devoid of nuclei and that (ii) blood mtDNA-CN does not correlate well with mtDNA-CN measured in other tissues in which mitochondrial dysfunction may be more relevant (Picard, 2021). Accordingly, understanding the genetic determinants of blood mtDNA-CN may provide a better understanding of the etiological processes reflected by this poorly understood mitochondrial biomarker in the blood.

To interrogate mtDNA-CN as a potential determinant of human diseases and to better understand its biological relevance to mitochondria, we performed extensive genetic investigations in up to 395,781 participants from the UKBiobank study(Sudlow et al., 2015).”

2. Page 9, lines 16-17: "We postulated that mtDNA-CN loci may regulate copy number by inducing changes in expression of genes that are directly transcribed from mtDNA.": I don't understand this rationale – are you postulating that lower/higher expression from the MT genome may trigger changes in copy number (and thus these may be modulated by variants influencing expression)? If so, this should be stated more clearly, if not, please explain the thinking here. I also think that it is worth rewording your conclusions from this analysis, as there is no test of directionality.

We apologize for the confusion here and have rephrased the rationale more clearly (page 9, lines 38-40):

“We postulated that differential expression of genes encoded by the mitochondrial genome may trigger changes in copy number, and thus a subset of identified loci may affect mtDNA-CN through mitochondrial genome transcription.”

3. Page 9: lines 31-41: Similar to point 2, it would be good to be a clear rationale for the comparison of loci associated with mtDNA-CN and heteroplasmy level.

Thank you for this comment. For further clarity, we have added a rationale for these analyses (page 10, lines 14-20):

“Heteroplasmy refers to the coexistence of multiple mtDNA alleles within an individual for a particular variant, which is a function of the multicopy nature of the mitochondrial genome. A recent GWAS by Nandakumar et al. (2021) for mean heteroplasmy levels in saliva specimen provided initial evidence supporting a shared genetic basis for heteroplasmy and copy number. To further explore the overlap in genetic determinants of these traits, we searched for the previously reported heteroplasmy loci within our mtDNA-CN GWAS.”

4. Page 10: lines 29-35: It would be good to see statistical tests comparing the fractions of genes that fall into the mitochondrial process categories described here, versus proportions of genes in these categories in Mitocarta as a whole – are these categories enriched for genes identified in this study?

The reviewer raises an excellent point and in response, we have performed enrichment analyses comparing the observed frequency of process categories to the null expectation based on the broader database of 1120 nuclear genes. Using a binomial test, we observe a significant enrichment in mitochondrial central dogma genes and have integrated these additions into the Results section (page 11, lines 19-24) accordingly:

“Most (16; 57%) genes were members of the “Mitochondrial central dogma” pathway, which represents a nearly 3-fold enrichment as compared to the frequency of this pathway in the whole MitoCarta3 database (null expectation = 20.7%; P=1.3x10^-5^). Other implicated (albeit not significantly enriched) pathways included “Metabolism”, “Mitochondrial dynamics and surveillance”, “Oxidative phosphorylation”, and “Protein import, sorting and homeostasis” (Figure 3D).”

5. Page 25: lines 20-31: Links should be provided in any revised manuscript, including the GitHub repository for the method, and summary statistics should be uploaded to the GWAS catalogue.

Thank you very much for this suggestion which was relayed by the editorial team as an essential revision. Accordingly, please see our detailed response above to essential revisions comment #4.

6. Figures: In general, the figures are not particularly clear and should be improved for publication. For example, axes text is often too small and difficult to read, and text labels on figures 1 and 2 are not aligned properly.

Thank you for this suggestion – we agree that the figure quality could have been improved. Accordingly, we provided higher resolution images for all figures, increased axes text size, and re-aligned text labels for Figures 1 and 2 (now Figures 3 and 4).